# Genomic epidemiology of the first two waves of SARS-CoV-2 in Canada

Angela McLaughlin[1,2]*, Vincent Montoya[1], Rachel L Miller[1,2], Gideon J Mordecai[3], Canadian COVID-19 Genomics Network (CanCOGen) Consortium[‡], Michael Worobey[4], Art FY Poon[5], Jeffrey B Joy[1,2,3]*

[1]British Columbia Centre for Excellence in HIV/AIDS, Vancouver, Canada; [2]Bioinformatics, University of British Columbia, Vancouver, Canada; [3]Department of Medicine, University of British Columbia, Vancouver, Canada; [4]Department of Ecology and Evolution, University of Arizona, Tucson, United States; [5]Department of Pathology and Laboratory Medicine, Western University, London, Canada

**Abstract** Tracking the emergence and spread of SARS-CoV-2 lineages using phylogenetics has proven critical to inform the timing and stringency of COVID-19 public health interventions. We investigated the effectiveness of international travel restrictions at reducing SARS-CoV-2 importations and transmission in Canada in the first two waves of 2020 and early 2021. Maximum likelihood phylogenetic trees were used to infer viruses' geographic origins, enabling identification of 2263 (95% confidence interval: 2159–2366) introductions, including 680 (658–703) Canadian sublineages, which are international introductions resulting in sampled Canadian descendants, and 1582 (1501–1663) singletons, introductions with no sampled descendants. Of the sublineages seeded during the first wave, 49% (46–52%) originated from the USA and were primarily introduced into Quebec (39%) and Ontario (36%), while in the second wave, the USA was still the predominant source (43%), alongside a larger contribution from India (16%) and the UK (7%). Following implementation of restrictions on the entry of foreign nationals on 21 March 2020, importations declined from 58.5 (50.4–66.5) sublineages per week to 10.3-fold (8.3–15.0) lower within 4 weeks. Despite the drastic reduction in viral importations following travel restrictions, newly seeded sublineages in summer and fall 2020 contributed to the persistence of COVID-19 cases in the second wave, highlighting the importance of sustained interventions to reduce transmission. Importations rebounded further in November, bringing newly emergent variants of concern (VOCs). By the end of February 2021, there had been an estimated 30 (19–41) B.1.1.7 sublineages imported into Canada, which increasingly displaced previously circulating sublineages by the end of the second wave.Although viral importations are nearly inevitable when global prevalence is high, with fewer importations there are fewer opportunities for novel variants to spark outbreaks or outcompete previously circulating lineages.

## Editor's evaluation

This study provides important observations about the transmission of SARS-CoV-2 lineages within Canada and the importation of lineages into Canada over the first year of the COVID-19 pandemic. This information is critical for understanding SARS-CoV-2 evolution and epidemiology, including the potential impacts of travel restrictions.

## Introduction

The COVID-19 pandemic has highlighted the importance of genomic epidemiology in deciphering the origin and spread of SARS-CoV-2 lineages across local and global scales to aid in directing responses

*For correspondence: amclaughlin@bccfe.ca (AMcL); jjoy@bccfe.ca (JBJ)

[‡]See supplementary materials for list of consortium members and affiliations.

(*Deng et al., 2020*; *Geoghegan et al., 2020*; *Gonzalez-Reiche et al., 2020*; *du Plessis et al., 2021*; *Worobey et al., 2020*). Studying evolutionary relationships between SARS-CoV-2 sequences over time and space allows estimation of the relative contributions of international and domestic transmission in association with public health interventions. Thus, phylogenetics can be highly informative in evaluating the effectiveness of non-pharmaceutical interventions at curbing importations and reducing transmission.

Phylogenetic analyses are dependent upon timely generation and sharing of publicly available genetic sequences and associated metadata, such as through the Global Initiative on Sharing All Influenza Data (GISAID) platform (*Khare et al., 2021*; *Shu and McCauley, 2017*). The global availability of SARS-CoV-2 genomes has been unprecedented, such that by 15 June 2021, just over a year and a half since the first whole genome sequence was shared (*de Maio, 2020*; *Wu et al., 2020*), there were 495,159 SARS-CoV-2 sequences on GISAID representing 108,301,802 COVID-19 diagnoses (*Krispin and Byrnes, 2020*). Nomenclature systems to partition viral sequences sharing common mutations and recent ancestry, such as Pango lineages, have provided a dynamic system for genomic SARS-CoV-2 surveillance (*O'Toole et al., 2021*; *Rambaut et al., 2020*). Tracking the emergence, dispersal, and genomic characteristics of lineages, particularly variants of concern (VOCs) and interest (VOIs), has become critical in light of their demonstrated increased transmissibility, potential increased clinical severity, and ability to circumvent host immune responses (*Lopez Bernal et al., 2021*; *Faria et al., 2021*; *Meng et al., 2021*; *Planas et al., 2021*; *Tegally et al., 2020*). Therefore, identifying and characterizing predominantly circulating lineages and sublineages is a cornerstone of global epidemiological surveillance and policy.

Phylogeographic methods to infer sampled viruses' dispersal have been widely applied to quantify SARS-CoV-2 introductions into the UK (*du Plessis et al., 2021*), the USA (*Deng et al., 2020*; *Gonzalez-Reiche et al., 2020*; *Worobey et al., 2020*; *Zeller et al., 2021*), Brazil (*Candido et al., 2020*), New Zealand (NZ) (*Douglas et al., 2021*; *Geoghegan et al., 2020*), and Europe (*Hodcroft et al., 2021*; *Worobey et al., 2020*), among others, elucidating variable epidemic dynamics. A recent review expounds further on the approaches and application of phylodynamic models towards improving our understanding of SARS-CoV-2 transmission and control (*Attwood et al., 2022*). In the UK, where the most sequences have been generated per case globally (*Furuse, 2021*), there were an estimated 1179 introductions resulting in two or more sampled descendant cases in the early epidemic (*du Plessis et al., 2021*). By contrast, the first wave of COVID-19 in Louisiana, USA, was primarily attributable to a single domestic introduction several weeks prior to Mardi Gras, resulting in multiple superspreader events and wide dissemination across Southern USA (*Zeller et al., 2021*). In NZ, where stringent border closures and lockdown measures were enacted early on, there were estimated to have been 277 introductions up to 1 July 2020, among which 19% resulted in multiple downstream cases (*Geoghegan et al., 2020*). Large-scale SARS-CoV-2 genomic epidemiology analyses in Canada have thus far been limited to a study on the early epidemic in the province of Quebec in which they conservatively estimated at least 500 viral introductions into Quebec by June 2020 largely attributable to the province's spring break (*Murall et al., 2021*). Our analyses elaborate upon their findings at a national scale for the first and second waves of COVID-19 in Canada.

Characterization of viral importations over time can also clarify the effectiveness of public health interventions by associating inflection points in importations with drastic changes in policies such as international travel restrictions (*Magalis et al., 2020*). Compartmental modelling approaches have also been applied to quantify the impact of COVID-19 control measures such as social distancing, informing decisions about when to relax or increase stringency (*Anderson et al., 2020*; *Anderson et al., 2021*). Evaluating the effectiveness of these interventions is key to reacting proportionally to the risk posed by future outbreaks of SARS-CoV-2 and other zoonotic pathogens. It is unclear how changes in intervention stringency, social and travel behaviour, and circulating viral diversity affected Canadian SARS-CoV-2 transmission dynamics, particularly during a time of negligibly low immunity prior to vaccine roll-out or widespread natural infections.

The first COVID-19 case in Canada was detected on 25 January 2020 in a traveller from Wuhan to Toronto and by 5 March, the first community transmission case was identified (*Press, 2021*; *Figure 1C*). In subsequent weeks, the stringency of Canadian interventions increased rapidly, summarized by the Oxford stringency index (*Hale et al., 2021*). On 14 March, a travel advisory warning against all non-essential travel outside Canada was issued; on 18 March, travel restrictions on the entry of all foreign

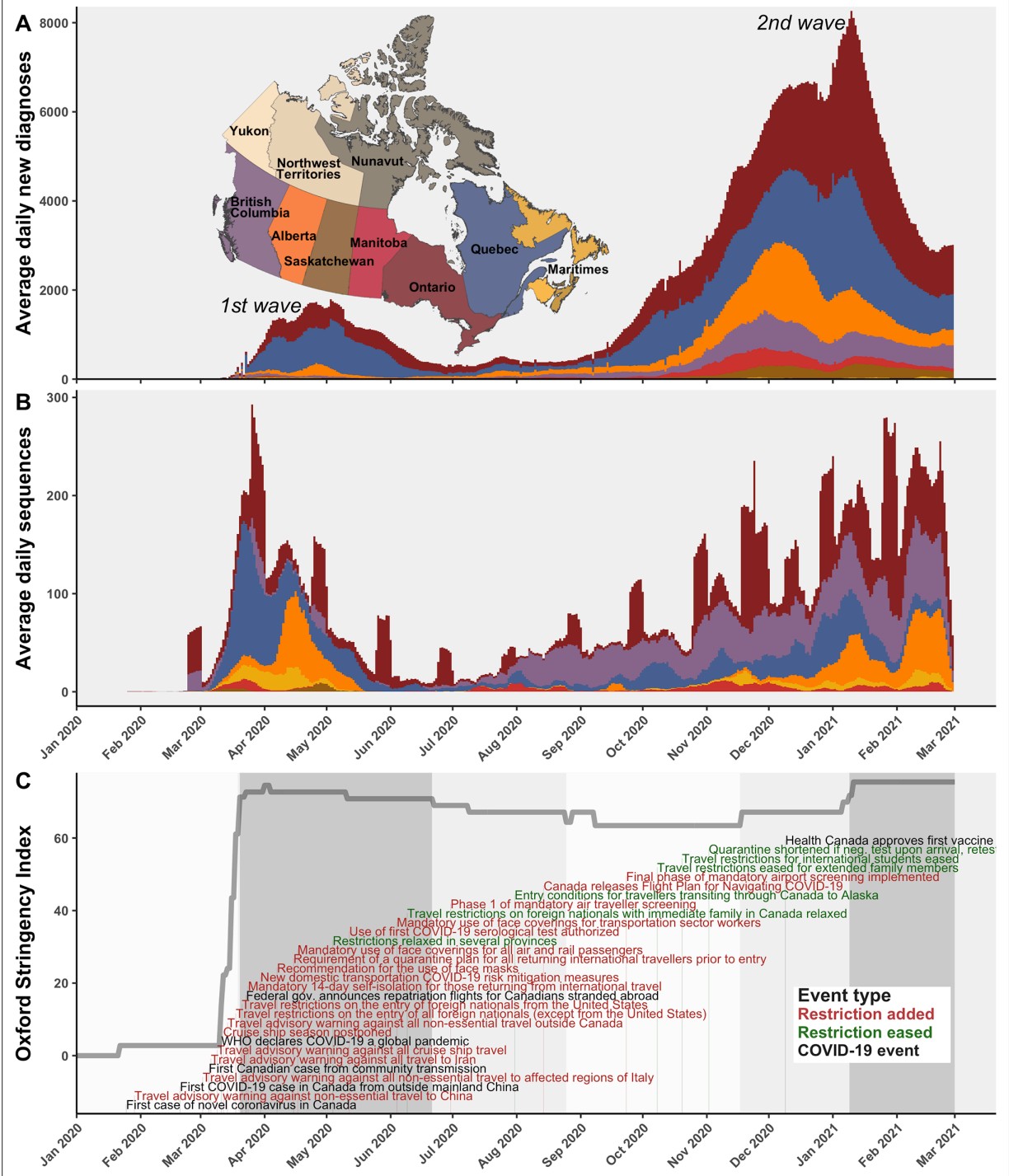

**Figure 1.** A timeline of the first and second waves of the Canadian COVID-19 epidemic up to 1 March 2021. (**A**) Rolling average daily new COVID-19 diagnoses in previous 7 days across Canadian provinces and territories, or the daily count where data was sparse prior to April 2020. (**B**) Rolling average daily clean SARS-CoV-2 sequences collected in previous 7 days, or daily sequences where data was sparse, uploaded to Global Initiative on Sharing All Influenza Data (GISAID) by 15 June 2021. Incomplete sample collection dates were inferred within time-scaled phylogenies. (**C**) The Oxford Stringency Index for Canada overlaid with key epidemiologic events and national-level public health restrictions. *Figure 1—figure supplement 1* summarizes the frequencies of Pango lineages among daily Canadian sequences. *Figure 1—figure supplement 2* compares monthly cases, sequences available, and sequences sampled globally and within Canada.

The online version of this article includes the following figure supplement(s) for figure 1:

**Figure supplement 1.** Canadian sequences available on Global Initiative on Sharing All Influenza Data (GISAID) over time by Pango lineage.

**Figure supplement 2.** Canadian and global sequences were subsampled with probabilities proportional to total monthly cases.

nationals (except from the USA) were implemented; 21 March travel restrictions were extended to the USA; and 24 March, a mandatory 14-day self-isolation for those returning from international travel was implemented. The stringency index during the first wave increased most rapidly on 16 March 2020, reaching the highest stringency between 1 and 3 April 2020. The first wave reached a maximum of 1108 average daily new COVID-19 diagnoses across Canada on 3 May 2020. While many of the earliest cases in Canada were attributable to A.1 and B.4 lineages in British Columbia (*Figure 1— figure supplement 1*), cases in March through June 2020 were primarily attributable to B.1* lineages and B.1.1* lineages in Ontario and Quebec, as well as lineage B.1.279 in Alberta in March and April. By June, in response to reductions in daily new diagnoses country-wide, restrictions slowly relaxed, resulting in a reduction of the stringency index, albeit with varying rates by province (*Cameron-Blake et al., 2021*). Fewer than 186 average daily new diagnoses occurred across Canada throughout summer 2020, mostly in Quebec, Ontario, Alberta, and British Columbia. Then, with the rising second wave in fall 2020, the stringency index rebounded in November 2020, although with simultaneous easing of entry exceptions for foreign nationals and quarantine shortening (*Cameron-Blake et al., 2021*). The second wave entered exponential growth in November 2020, as VOCs and VOIs began to displace wild-type lineages.

The VOCs and VOIs first detected in Canada were B.1.1.7 (Alpha) on 8 November 2020, followed by P.2 on 24 November, A.23.1 on 2 December, B.1.429 on 5 December, B.1.525 (Eta) on 14 December, B.1.351 (Beta) on 19 December, B.1.427 (Epsilon) on 30 December (dates from GISAID). The winter holidays were followed by the crest of the second wave on 11 January 2021 with 3555 average daily new diagnoses in Canada, likely dampened by restrictions on travel, personal gatherings, dining, and mask use. The new year also brought first Canadian detections of B.1.526.1/.2 (Iota) on 1 January 2021, P.1 (Gamma) on 1 February, and C.37 (Lambda) on 15 February. By March 2021, the trough of the second wave gave heed to the third wave, driven almost exclusively by B.1.1.7 and P.1, the latter in Western provinces primarily. Monitoring lineage frequencies alone does not inform how many individual importations accounted for the detected cases, nor the domestic spread dynamics of individual sublineages, warranting a genomic epidemiology analysis.

A phylogenetic maximum likelihood approach was applied to estimate the timing, origins, destinations, and transmission of SARS-CoV-2 introductions in Canada from the beginning of 2020 to the end of the second wave in March 2021 for Canadian sublineages (introductions with sampled descendants) and singletons (introductions with no sampled descendants). We tested the hypothesis that international travel restrictions enacted in March 2020 effectively reduced international importations of SARS-CoV-2 into Canada, yet ongoing introductions contributed to COVID-19 persistence into early 2021, exacerbated by highly transmissible B.1.1.7 and other VOC sublineages. These analyses help to elucidate the relative contributions made by sublineages introduced before and after enactment of travel restrictions towards persistence of the Canadian SARS-CoV-2 epidemic in 2020 and early 2021, prior to the global predominance of VOCs and vaccine roll-out. Evaluating travel restrictions' effectiveness towards reducing importations could inform the stringency and timing of future public health interventions.

## Results
### Global SARS-CoV-2 phylogeny with a Canadian focus

The overrepresentation of sequences from countries with the highest sequencing efforts, such as the UK, was reduced in the dataset by subsampling (*Figure 1—figure supplement 2*; *Figure 6—figure supplement 2*). Cumulative sequence representation was increasingly comparable across provinces with fewer Canadian sequences subsampled, however excluding too many sequences resulted in missed introductions. There were no sequences available from the Yukon, Northwest Territories, Nunavut, or Prince Edward Island, which cumulatively had 603 COVID-19 cases (0.07% of Canada cases) prior to 1 March 2021.

Canadian viral genomes were dispersed throughout the global phylogeny and represented 345 unique Pango lineages (*Figure 2*). The average time to the most recent common ancestor (TMRCA) was estimated as 22 December 2019 (24 November–23 December) across 10 subsamples, consistent with the upper end of others' credibility intervals (*Lu et al., 2020*; *Worobey et al., 2020*). The relaxed molecular clock model estimated in LSD2 had a mean rate of $2.86 \times 10^{-4}$ ($2.84 \times 10^{-4}$–$2.92 \times 10^{-4}$)

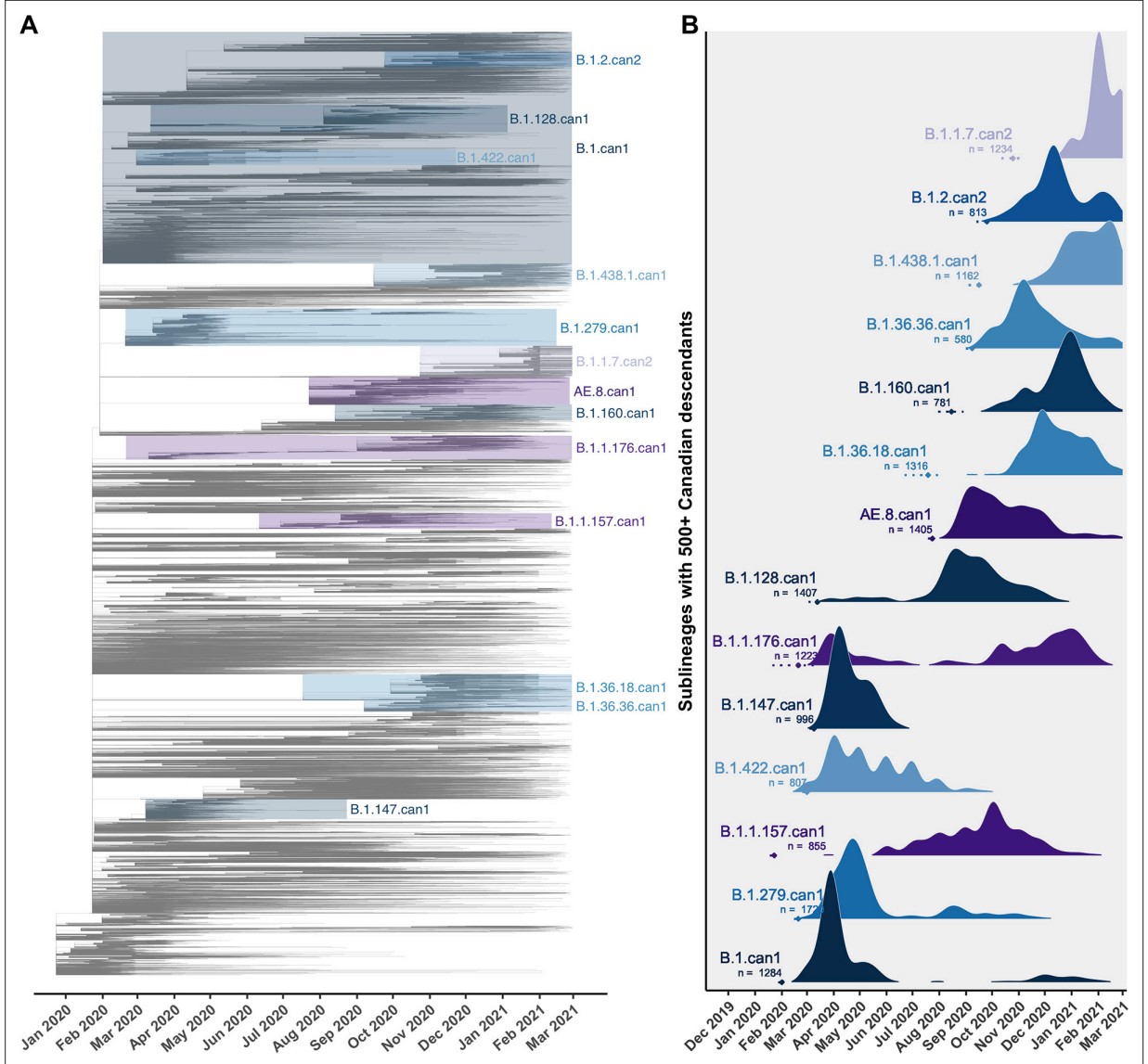

**Figure 2.** Key Canadian sublineages in a phylogenetic tree of SARS-CoV-2 in Canada and globally up to the end of the second wave on 28 February 2021. (**A**) The highest likelihood bootstrap time-scaled phylogenetic tree inferred using a subsampling strategy where 75% of available Canadian sequences were retained and the remainder of sequences up to 50,000 were from global sources. (**B**) The timing and expansion of key Canadian SARS-CoV-2 sublineages with more than 500 sampled Canadian descendants. Height reflects the relative density of sampled Canadian descendants within each sublineage. Diamonds and dashed lines show the mean and 95% confidence interval of the time to the most recent common ancestor (TMRCA). *Figure 2—figure supplement 1* zooms in on the subtrees for the four largest Canadian sublineages. *Figure 2—figure supplement 2* summarizes the process of removing temporal outliers to improve molecular clock signal. *Figure 2—figure supplement 3* summarizes sublineage introduction node and parent node likelihoods overlaid for all bootstraps.

The online version of this article includes the following figure supplement(s) for figure 2:

**Figure supplement 1.** The four Canadian sublineages with the most sampled Canadian descendants by 28 February 2021.

**Figure supplement 2.** Removal of temporal outliers to improve the molecular clock rate fit.

**Figure supplement 3.** Sublineage introduction node and parent node likelihoods overlaid for all bootstraps in the 75% subsampling strategy, annotated with the percent of all Canadian sublineage-defining introductory nodes across bootstraps within each threshold.

substitutions/site/year (s/s/y), lower than the mean strict clock rate inferred in TempEst of $7.15 \times 10^{-4}$ ($6.83 \times 10^{-4}$–$7.48 \times 10^{-4}$) s/s/y (*Figure 2—figure supplement 2*) due to differing assumptions of rate variation. Increasing the representation of global samples relative to Canadian sequences generally resulted in earlier TMRCA estimates, higher estimated clock rates (sampling wider diversity), more

temporal outliers removed, and lower $R^2$ values associated with their clock models (**Supplementary file 3c**; **Supplementary file 3d**). The large majority of sublineages (75.1%) were supported by high likelihoods (>0.9) in both introduction and parental nodes across all bootstraps (**Figure 2—figure supplement 3**), consistent across subsampling strategies. The 14 sublineages with 500 or more Canadian descendants cumulatively accounted for 59.1% of Canadian sublineage descendants, despite representing only 2.3% of sublineages (**Figure 2**).

## Diverse origins of Canadian SARS-CoV-2 sublineages

Maximum likelihood discrete ancestral state reconstruction of the time-scaled phylogenies with 75% of Canadian sequences yielded a mean of 680 (658–703) Canadian sublineages (introductions with sampled descendants) by the end of February 2021 (**Figure 3**) and 1582 (1501–1663) singletons (introductions with no sampled descendants) (**Figure 3—figure supplement 2**). Although singletons accounted for 70% (69–71%) of introductions, some were likely representative of sublineages with no other sampled descendants. The proportion of importations resulting in domestic transmission did not vary significantly by province, origin, or month (Kruskal-Wallis test: p=0.53, p=0.47, p=0.37), but notably reached a maximum of 0.94 in February 2020 (**Figure 6—figure supplement 3**).

We estimated that during the first wave, from 1 January to 31 July 2020, there were a total of 384 (368–400) introductions resulting in domestic transmission, including 169 (155–182) B.1 sublineages, 53 (50–56) B.1.1 sublineages, 23 (20–25) A.1 sublineages, 9 (8–10) B.4 sublineages, 8 (7–9) B.1.36 sublineages, and 7 (6–7) B.40 sublineages (**Figure 3A**). Most first wave Canadian sublineages originated from the USA (49%, 46–52%), followed by Russia (12%, 11–13%), Italy (6%, 5–9%), India (6%, 5–7%), Spain (5%, 4–5%), and the UK (4%, 4–5%), among others. These sublineages were inferred to have been primarily imported into Quebec (39%, 38–40%) followed by Ontario (36%, 35–38%), and British Columbia (12%, 11–13%), followed by Manitoba (5%, 5–5%), the Maritimes (5%, 5–5%), Alberta (2%, 2–2%), and Saskatchewan (1%, 0–1%). The second wave, from 1 August 2020 to 28 February 2021, included 296 (280–313) newly introduced sublineages, primarily of B.1.2 (55, 54–56), B.1.1.7 (30, 19–40), B.1.36 (24, 22-25), B.1 (14, 12–15), and B.1.1.519 (8, 8–9) lineages (**Figure 3B**). The origins of sublineages introduced during the second wave remained predominantly from the USA (43%, 40–45%), alongside increased relative contributions from India (16, 15–17%), the UK (7%, 5–9%), Asia (6%, 5–6%), Europe (4%, 4–5%), and Africa (3%, 3–4%). There were relatively more sublineages introduced to Ontario (45%, 44–47%), British Columbia (23%, 22–24%), and Alberta (11%, 10–11%) during the second wave, and fewer to Quebec (16%, 16–17%) than in the first wave. Relative contributions of origins and destinations were mostly robust to subsampling strategy, although differed in their absolute estimates (**Figure 6—figure supplement 4**).

Sublineage sizes were overdispersed (**Figure 3—figure supplement 2**) with a mean of 53 (47–59) total unique sampled descendants (globally and within Canada) and a median of 4 (4–4). There were 46 (44–49) sublineages with more than 100 unique sampled descendants. Together, they represented 6.8% (6.4–7.2%) of sublineages and 0.7% (0.6–0.8%) of total introductions including singletons, yet their Canadian descendants accounted for 77% (74–81%) of Canadian sequences. Only 14 sublineages had 500 or more sampled Canadian descendants (**Figure 2**). The largest sublineages from the first wave, including B.1.279.can1, B.1.can1, B.1.128.can1, and B.1.1.176.can1, persisted through to the second wave (**Figure 2—figure supplement 1**), although their descendants no longer accounted for the majority of Canadian samples (**Figure 3D**). In the second wave, the most frequently sampled sublineages in Canada were AE.8.can1 (introduced in July 2020), B.1.36.18.can1 (July 2020), B.1.1.7.can1 (November 2020), and B.1.438.1.can1 (September 2020). As the second wave progressed into the new year, increasingly VOC sublineages, B.1.1.7.can2 and B.1.1.7.can3, outcompeted previously dominant sublineages, B.1.36.18.can1, B.1.128.can1, and AE.8.can1 (**Figure 3—figure supplement 4**). A total of 34,169 (31,337–37,001) sampled descendants of Canadian sublineages were identified (**Figure 3—figure supplement 3**). While the majority of descendants were in Canada (n=25,270, 25,071–26,369), the 8449 (5915–10,983) global descendants were vastly underestimated by subsampling.

## Travel restrictions reduced sublineage importations

Sublineages' TMRCA approximates the date of the first transmission event resulting in one or more sampled descendant following viral introduction. Although TMRCA is not equivalent to the date of a

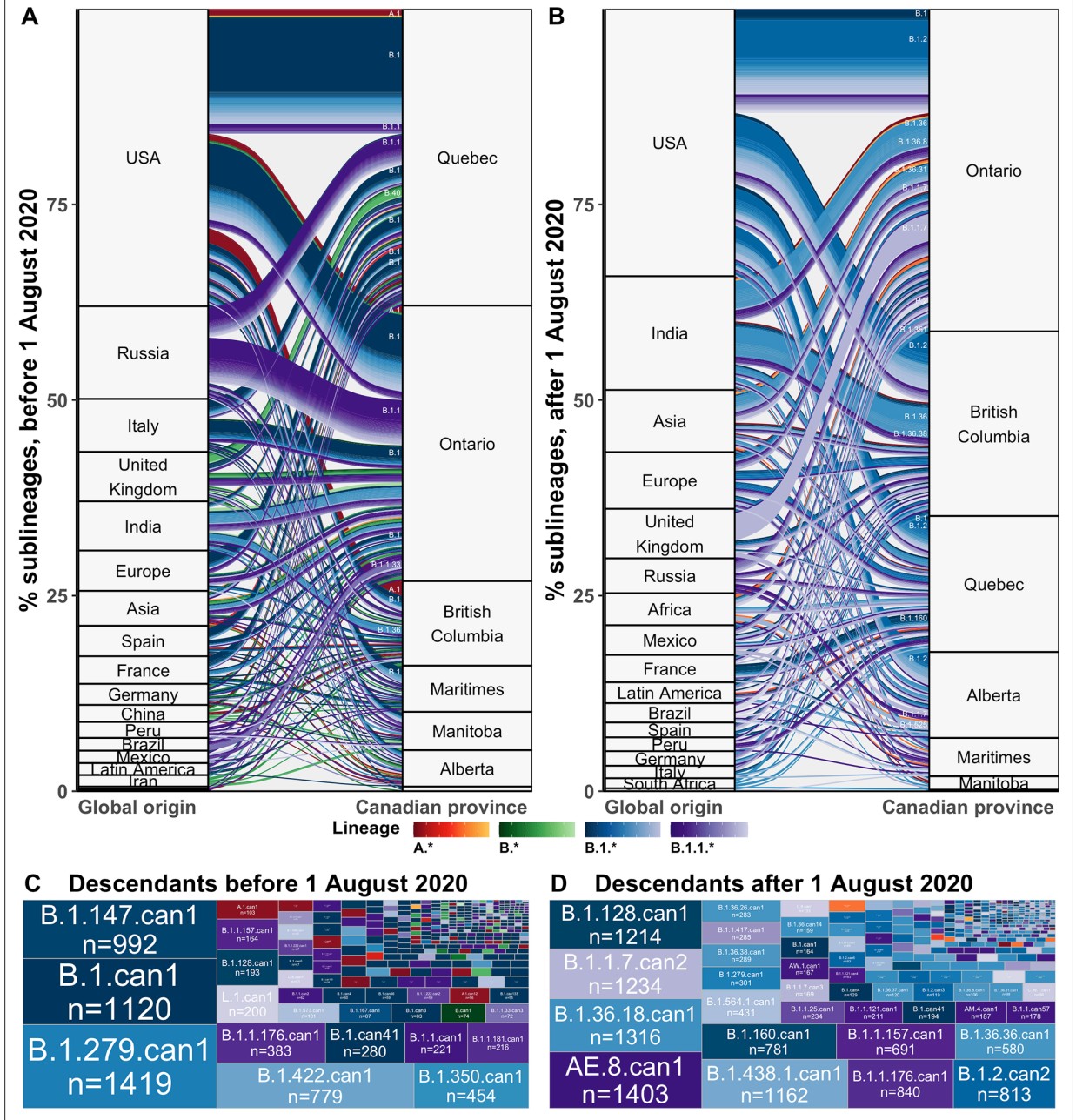

**Figure 3.** Flow of SARS-CoV-2 sublineages into Canadian provinces from global origins coloured by Pango lineage in (**A**) the first wave, before 1 August 2020, and (**B**) the second wave, from 1 August 2020 to 28 February 2021. Locations' relative size and lineage flows between location pairs represent the mean percent of sublineages across 10 subsamples for the 75% sampling strategy. Lineages by location pair associated with greater than 0.5% of sublineages and locations associated with more than 1% of sublineages were labelled. Sublineages' relative sizes, based on the number of sampled unique Canadian descendants during the (**C**) first and (**D**) second wave. Canadian sublineages were named based on the predominant Pango lineage of descendants and a 'can' suffix with a numeric denoting their order of first Canadian sample date. *Figure 3—figure supplement 1* shows the flow of singletons. *Figure 3—figure supplement 2* summarizes the distribution of sublineage sizes. *Figure 3—figure supplement 3* shows the distribution of all Canadian sublineages' descendants' sampling locations. *Figure 3—figure supplement 4* stratifies second wave sublineage sizes by pre- and post-January 2021. *Figure 3—figure supplement 5* stratifies sublineages' overall size by global and Canadian descendants.

The online version of this article includes the following figure supplement(s) for figure 3:

**Figure supplement 1.** The flow of singletons from global origins to Canadian provinces by Pango lineage in the (**A**) first and (**B**) second waves.

**Figure supplement 2.** The distribution of sublineage sizes including Canadian and global descendants representing the highest likelihood bootstrap tree from the 75% Canadian sequences retained sampling strategy.

*Figure 3 continued on next page*

*Figure 3 continued*

**Figure supplement 3.** Distribution of sampling locations for descendants of Canadian SARS-CoV-2 sublineages.

**Figure supplement 4.** Canadian SARS-CoV-2 sublineages' relative sizes, based on the number of sampled Canadian descendants during (**A**) the first half of the second wave, from 1 August to 31 December 2020, and (**B**) the second half of the second wave, after 1 January 2021.

**Figure supplement 5.** The relative sizes of Canadian SARS-CoV-2 sublineages across both waves when considering (**A**) all descendants, sampled globally and within Canada, and (**B**) only Canadian descendants.

viral host crossing a border, they are associated by a sampling lag and possibly one or more unsampled generations.

By tracking sublineages' TMRCA over time, the temporal dynamics of SARS-CoV-2 importations were modelled by global origin and Canadian destination (*Figure 4*). There may have been upwards of 14 (7–21) sublineages introduced prior to the first COVID-19 case in Canada identified on 25 January 2020, and 233 (218–248) sublineages prior to the implementation of travel restrictions for all foreign nationals on 21 March 2020. However, early sublineage introduction dates have wide uncertainty due to low phylogenetic diversity and large polytomies. On the day following imposition of travel restrictions (22 April 2020), the importation rate reached its maximum of 58.5 (50.4–66.5) sublineages per week, including 31.8 (27.7–35.9) sublineages per week originating from the USA and 31.2 (28.7–33.7) sublineages per week introduced into Quebec. Ontario had reached its maximum several days before

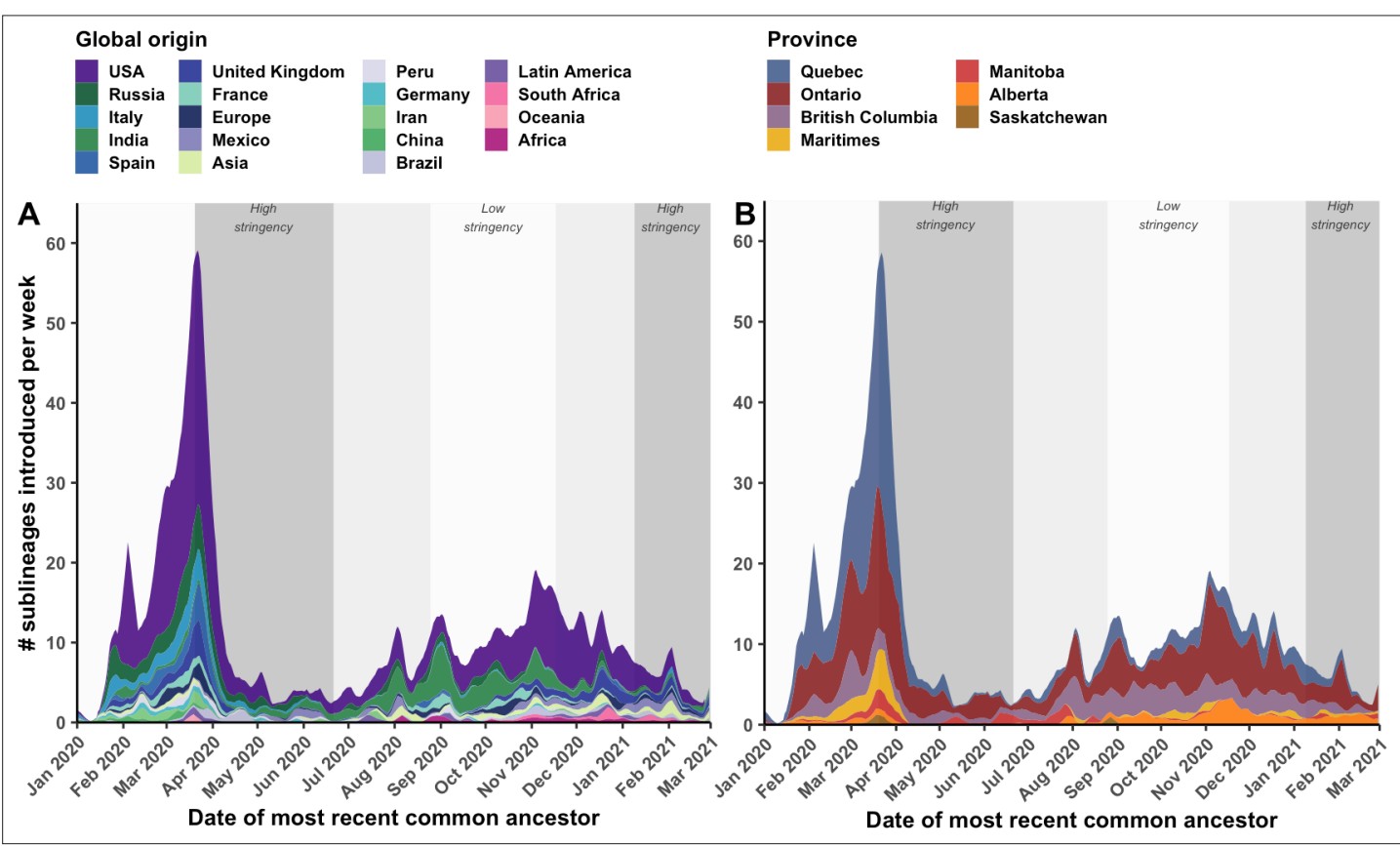

**Figure 4.** Weekly introduction rates of Canadian SARS-CoV-2 sublineages in the first two waves, in the context of changes in COVID-19 intervention stringency. The weekly sublineage introduction rates were summarized as 7-day rolling means across bootstraps (**A**) by global origin and (**B**) by province of introduction. The background shading corresponds to periods of high and low stringency, based on Oxford COVID-19 Stringency Index. *Figure 4—figure supplement 1* summarizes weekly singleton introduction rates. *Figure 4—figure supplement 2* characterizes changes in sublineages' size, detection lag, and lifespan over time.

The online version of this article includes the following figure supplement(s) for figure 4:

**Figure supplement 1.** Weekly singleton introduction rates over time.

**Figure supplement 2.** Characterization of Canadian SARS-CoV-2 sublineages over time.

on 19 March, culminating at 17.7 (15.0–20.5) sublineages per week. Over this period, many Canadians were repatriated from around the world, yet the mandatory 14-day at-home quarantine was not enacted until 25 March. By 5 April, 2 weeks after travel restrictions took effect, the overall sublineage importation rate had dropped 3.4-fold (3.2–3.8); and within 4 weeks, the rate had dropped 10.3-fold (8.3–15.0). This inferred inflection point in sublineage importation rate was robust to multiple subsampling strategies (*Figure 6—figure supplement 5*). The extent to which importations were reduced within 2 weeks of heightened stringency varied by province; importations were reduced by 9.9-fold (1.3–18.6) in the Maritimes, 5.3-fold (4.3–6.2) in Quebec, 2.2-fold (1.7–2.7) in Ontario, 1.6-fold (0.8–2.3) in British Columbia, and 0.94-fold (0.2–1.7) in Alberta. Importation dynamics for singletons mostly mirrored sublineage trends (*Figure 4—figure supplement 1*; *Figure 3—figure supplement 1*), although the USA contributed relatively more towards singletons than sublineages in both waves, and the maximum singleton importation rate was higher in the first wave than the second wave.

Despite reductions, introductions were maintained at a low level (1.0–12.5 sublineages per week) until 1 August 2020, when a small spike in importations was detected leading into the second wave. Upon further relaxation of travel restrictions for incoming international students and Canadians' family members in mid-October 2020 (*Canadian Institute for Health Information, 2021*), coinciding with a sustained low stringency, importation rates rebounded to 17.5 (13.5–21.5) sublineages per week on 2 November; then, with increased travel prior to the holidays, rates increased to (10.5–18.9) sublineages per week on 3 December. Prior to the new year, numerous VOC and VOI sublineages had been introduced to Canada, including 15 (10–21) B.1.1.7, 5 (4–5) B.1.351, 2 (1–2) B.1.429, 5 (4–5) P.2, and 4 (4-5) A.23.1 sublineages.

## Sublineage size, lifespan, and detection lag over time

First, to elucidate the relative contributions of early and late sublineages, sublineage size (number of sampled descendants) over time was evaluated (*Figure 4—figure supplement 2A*). Sublineages were stratified by whether they were active (had any sampled Canadian cases in past 2 months before 1 March 2021) or extinct, to account for differences in sublineages' time to accrue and sample cases. Sublineage size was significantly reduced over time among active and extinct sublineages (both p<0.001). Concurrently, sublineages' transmission lifespan (days from TMRCA to most recent Canadian sample) decreased over time for both active and extinct sublineages (both p<0.001, *Figure 4—figure supplement 2B*).

Under a null hypothesis where both earlier and later sublineages were equally likely to be transmitted and sampled, we would expect the mean number of days since importation to be steady over time. While the average number of days since importation of Canadian samples (days from sublineage TMRCA to sample date) steadily increased over time during the first wave, it dipped several times in summer 2020 as new sublineages became more widespread, leveling off the age since importation in the second wave (*Figure 4—figure supplement 2C*). Average sublineage detection lag (days from TMRCA to first Canadian sample) did not change significantly over time (p=0.62; *Figure 4—figure supplement 2D*). There was insufficient evidence that detection lag was significantly associated with province of introduction (Kruskal-Wallis test, p-value = 0.23).

## International, domestic, and provincial transmission sources

The inferred transmission source (most likely state at internal node directly preceding tip) of all sampled Canadian genomes was investigated to highlight the relative roles of domestic and international transmission by province over time. Transmission sources were categorized as within-province, between-province, the USA, or other international sources. Following the entry restriction for foreign nationals in mid-March, there was a reduction, but not an elimination, of the proportion of transmission events attributable to the USA or other international sources across all provinces (*Figure 5A*). Prior to the restrictions, provinces had a mean of 26% of transmission events with any international origins, ranging from 17% (14–20%) in Saskatchewan to 33% (31–35%) in British Columbia. In April 2020, following travel restrictions, all provinces had decreased proportions of international transmission sources, yet Ontario and Quebec retained relatively high mean number of transmission events attributable to international sources, at 115 (105–125) and 95 (82–109) (*Figure 5—figure supplement 1*). This may suggest slower implementation or compliance with quarantine guidelines in these provinces.

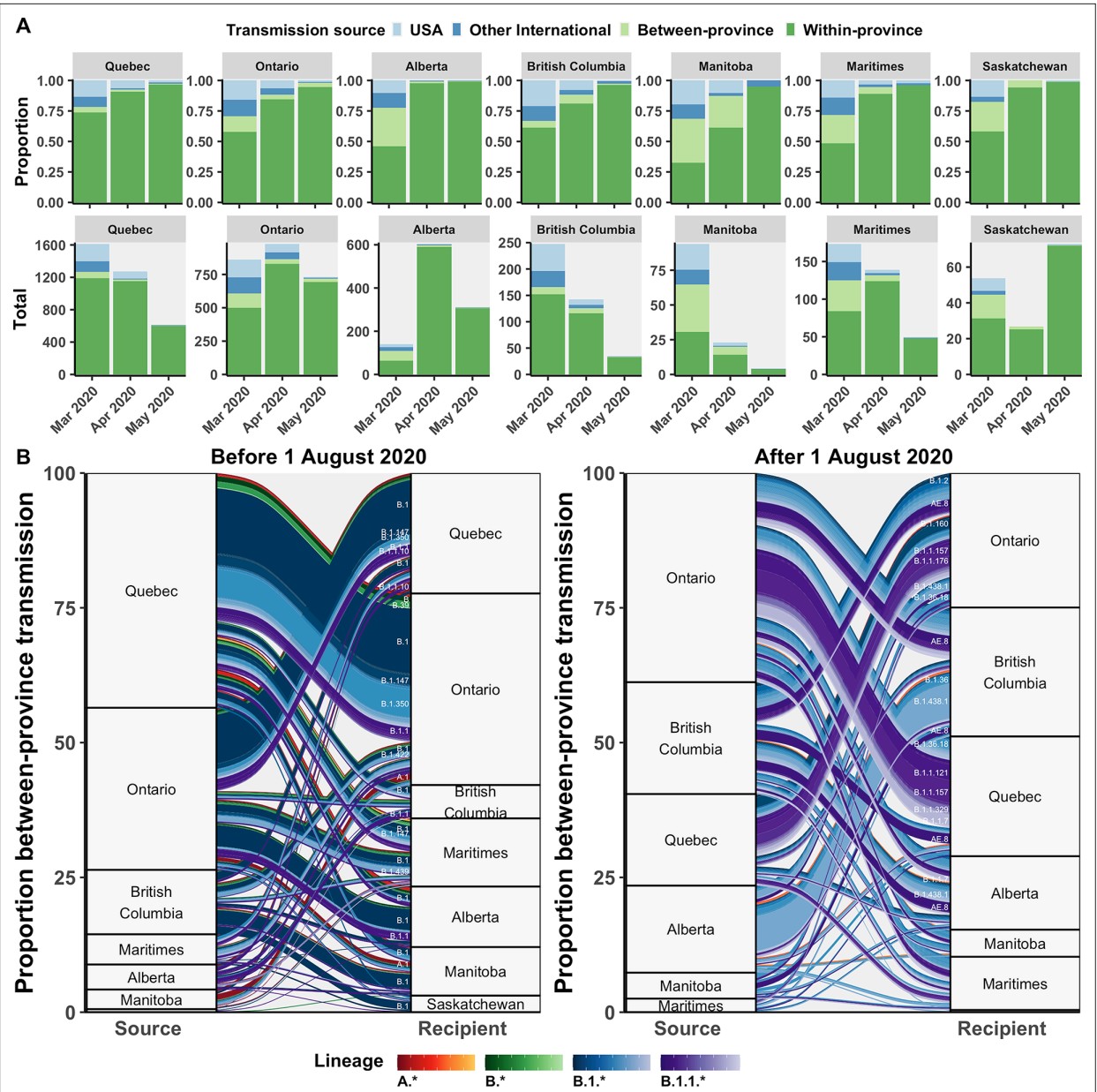

**Figure 5.** Relative contributions of international and domestic transmission sources. (**A**) Proportional and total contributions of the USA, other international, between-province, and within-province transmission sources among all sampled tips by province and month between in March, April, and May 2020. (**B**) The proportional flow of lineages transmitted between-provinces among Canadian tips, stratified before and after August 2020. Values reflect the mean across bootstraps for the 75% subsampling strategy. *Figure 5—figure supplement 1* depicts the total number of sampled transmission events with any international source across Canada in April 2020.

The online version of this article includes the following figure supplement(s) for figure 5:

**Figure supplement 1.** The mean number of sampled transmission events with an inferred international source across Canada in April 2020, immediately following the enactment of travel restrictions.

Between-province transmission was investigated by comparing transmission origins among Canadian sequences with an inferred Canadian origin from another province (*Figure 5B*). During the first wave, Ontario and Quebec were the greatest sampled sources of interprovincial transmission, with B.1, B.1.1, and B.1.350 accounting for the largest cumulative flows. In the second wave after August 2020, Ontario remained the greatest sources of interprovincial importations, followed by nearly equal contributions from British Columbia, Alberta, and Quebec. Dominant lineages transmitted interprovincially in the second wave were B.1.438.1 from Alberta to British Columbia, B.1.1.157 and B.1.1.121

from Ontario to Quebec, AE.8 from British Columbia to Quebec and Ontario to Quebec, and B.1.1.7 from Ontario to Quebec.

### Subsampling sensitivity analysis

To render phylogenetic inferences computationally feasible, reduce overrepresentation of geographies with more sequences per case, and evaluate uncertainty in the effect of sampling on importation rates, sequences were subsampled proportionally to either provinces' or countries' contributions to monthly case counts in Canada and globally for multiple bootstraps. We compared four subsampling strategies where 25% (n=9184), 50% (n=18,368), 75% (n=27,552), or 100% (n=36,736) of Canadian sequences were retained and global sequences were sampled up to 50,000 total. Each subsampling strategy was repeated for 10 bootstraps with replacement.

While the 75% strategy identified the most sublineages, it also had the widest 95% confidence interval surrounding the number of sublineages of all the strategies (*Figure 6*). The median 95% confidence interval widths for number of sublineages for each location pair was comparable across strategies, and each strategy included several large widths for high estimates. Some variation is expected within a strategy across bootstraps as the sampling process is stochastic, that is, countries were sampled probabilistically, rather than proportionally. The 75% strategy also identified the most singletons, followed by the 50%, 100%, and 25% strategies. When too few Canadian sequences were included, sublineages and singletons went undetected more often; however, when too few global sequences were included as a result of high inclusion of Canadian sequences (under soft computational limit of 50,000 sequences), migration events went undetected due to underrepresented global genetic diversity. Therefore, to maximize our ability to identify domestically circulating sublineages, the 75% strategy was reported in the main figures.

As the percent of Canadian sequences retained increased and the number of global sequences sampled each month decreased, there was decreased overrepresentation of sequences with strong sampling efforts including the UK and USA (*Figure 6—figure supplement 2*). The 75% Canadian sequences subsampled strategy achieved the highest Pearson's correlation coefficient between total cases and sequences for global regions (0.96), with a mean of 0.075 global sequences per 100 cases (*Figure 6—figure supplement 1*). Among Canadian provinces within the 75% strategy, the correlation coefficient was 0.87 with 4.5 sequences per 100 cases on average, which was surpassed by the strategies with fewer sequences retained. With fewer Canadian sequences sampled, provinces' contributions of sequences over time were more normalized. For all subsampling strategies, China and the USA had higher than average representation on account of their contributions during early pandemic months when sequences were sparse and all included in the analysis to best reconstruct early lineage divergence. Iran and Saskatchewan had consistently lower than average sequence representation.

The relative ranking of the mean number of sublineages global regions and Canadian provinces were mostly robust to subsampling strategy, although there were subtle differences, particularly for regions associated with fewer sublineages (*Figure 6—figure supplement 4*). Notably, with only 25% of Canadian sequences retained, the estimated percent of sublineages introduced to British Columbia in the second wave was less than Quebec, whereas in the 75% and 100% strategies, British Columbia accounted for more sublineages than Quebec, partially as a result of less normalization of sequence representation. Spatiotemporal trends in sublineage importation rates were comparable between sampling strategies, although the amplitude of importations during the first wave varied, with the 75% subsampling strategy displaying the largest first wave crest, attributable primarily to the USA (*Figure 6—figure supplement 5*). The second wave amplitude and trends were relatively similar between strategies. The drastic reduction of importation rates following the implementation of travel restrictions in late March was robust across strategies, with comparable inflection points. Across all strategies, there were significant reductions of sublineage size and lifespan over time.

## Discussion

Together, these analyses support the conclusion that travel restrictions and other non-pharmaceutical interventions imposed in March 2020 drastically reduced importations, preventing the expansion of subsequent sublineages. A more rapid and stringent public health response would have reduced the initial burden by preventing early sublineages from establishing widespread transmission chains that

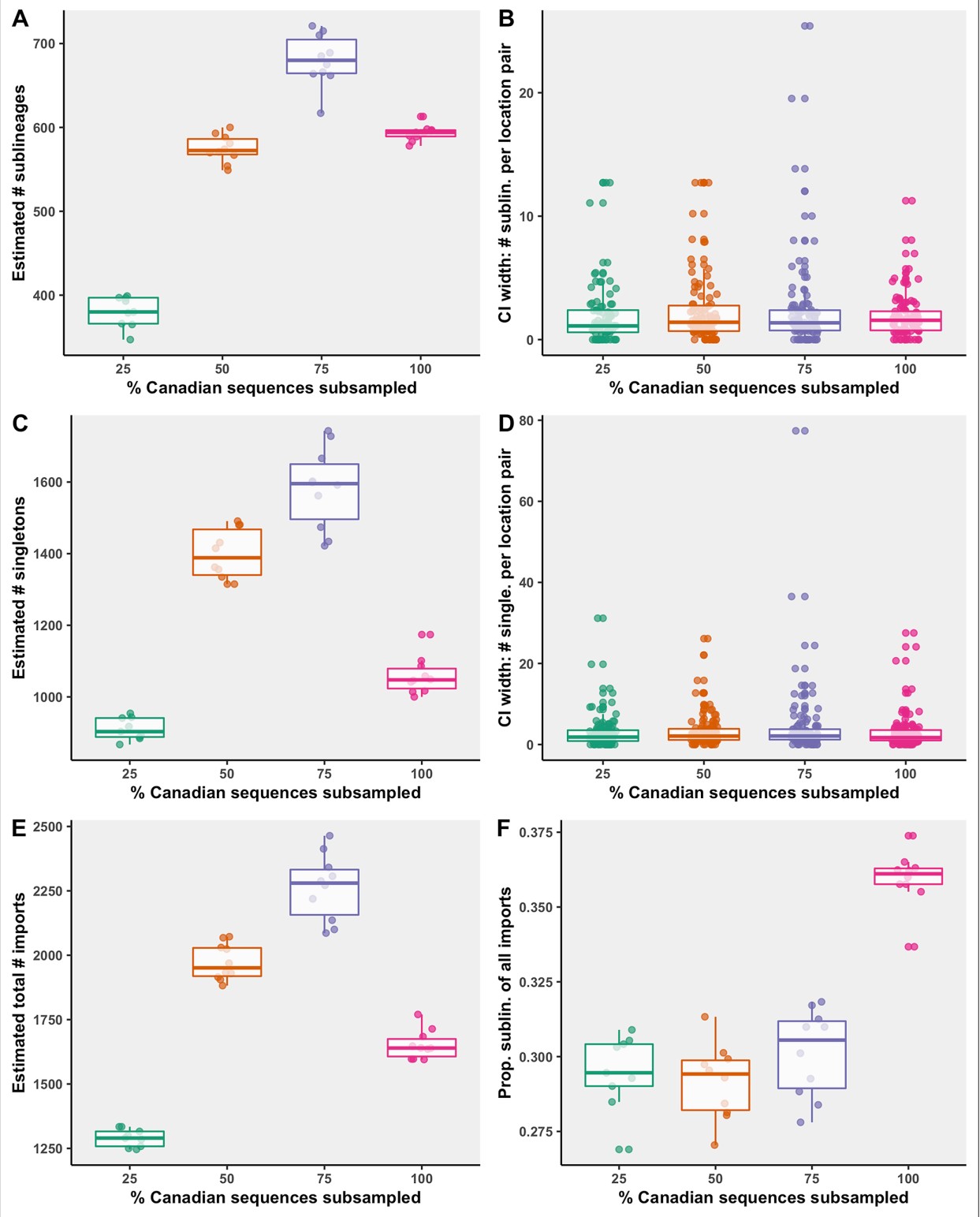

**Figure 6.** Subsampling sensitivity analysis, with 25–100% of Canadian sequences retained. (**A**) The estimated number of sublineages across 10 bootstraps. (**B**) The distribution of 95% confidence interval widths for number of sublineages attributable to each location pair across 10 bootstraps. (**C**) The estimated number of singletons identified across 10 bootstraps for each subsampling strategy. (**D**) The distribution of 95% confidence interval widths for number of singletons attributable to each location pair across 10 bootstraps. (**E**) The estimated total number of importations, that is, the sum of sublineages and singletons, and (**F**) the proportion of sublineages among all importations. Subsampling strategies are further compared in regard to the relationship between sequences and cases by strategy in *Figure 6—figure supplement 1*; subsampled sequence densities in *Figure 6—figure*

*Figure 6 continued on next page*

*Figure 6 continued*

*supplement 2*; mean proportion of importations resulting in a sublineage by month and region in *Figure 6—figure supplement 3*; relative sublineage contributions in *Figure 6—figure supplement 4*; and finally sublineage introduction rates in *Figure 6—figure supplement 5*.

The online version of this article includes the following figure supplement(s) for figure 6:

**Figure supplement 1.** The relationship between total clean sequences available and total cases across global regions (**A–E**) and Canadian provinces (**F–J**) for all months in the study period for each subsampling strategy and before subsampling.

**Figure supplement 2.** Subsampled sequence densities for global regions (**A–D**) and Canadian provinces (**E–H**) with 25–100% of Canadian available sequences retained.

**Figure supplement 3.** The mean proportion of importations resulting in a sublineage (versus a singleton), by month, province of introduction, and subsampling strategy.

**Figure supplement 4.** Comparative alluvial plots across subsampling strategies with 25–100% of Canadian sequences retained in the first and second waves (top and bottom rows).

**Figure supplement 5.** A comparison of the rolling sublineages introduced per week across subsampling strategies.

persisted throughout the first and second waves. However, wild-type (pre-VOC) sublineages introduced in summer 2020 when prevalence and immunity were low contributed the highest proportion of COVID-19 cases in the second wave, suggesting that even a low level of ongoing importations of similarly transmissible variants can contribute to viral persistence. Over time, improvements in quarantine, contact tracing, testing, and individual behavioural changes collectively contributed to smaller sublineages with shorter transmission lifespans, prior to the seeding of highly transmissible VOC sublineages. A moderate level of border porosity throughout 2020 and 2021 provided opportunities for importations of B.1.1.7 and other VOCs that outcompeted previously circulating sublineages to near extinction in the third wave.

Although the USA was highly represented in all of our subsamples as a result of its large contribution to COVID-19 cases in 2020 and high sequence availability during early months, our results suggest a greater effect than due to sampling alone. On average, the USA sequences represented 28.9% (28.7–29.2%) of total international sequences, yet accounted for 46.3% (44.0–48.7%) of all sublineages and 57.7% (55.6–59.8%) of singletons. Upon maximizing the number of Canadian sequences in the analysis, where global sequence representation was more normalized but less comprehensive, the USA sequences represented fewer of the international sequences (25.8%, 25.6–26.1%) and still accounted for 38.4% (37.0–39.8%) of sublineages and 46.4% (44.6–48.3%) of singletons. It was foreseeable that the USA would be the largest source of SARS-CoV-2 importations into Canada given its high COVID-19 prevalence throughout 2020 as well as the fact that these nations share the world's longest land border, spanning 8890 km. Overall international arrivals into Canada declined 77.8% from 96.8 M in 2019 to 19.7 M in 2020 (*Canada Statistics, 2021a*). However, the number of truck drivers and crew members (air, ship, and train), only declined by 24.8%, accounting for 49% of all international arrivals after April 2020 (of whom 93% were truck drivers). Truck drivers supported the supply chain throughout COVID-19 disruptions, yet in doing so, may have inadvertently facilitated additional SARS-CoV-2 importations from the USA. International arrivals of Canadians and non-residents in 2020 were low, but not negligible, with 14.6 M and 5.1 M arrivals respectively in 2020 (*Canada Statistics, 2021a*). While essential workers were exempt from quarantine, all others were mandated a self-enforced 14-day quarantine upon re-entry. It was not until 22 February 2021 that a 3-day hotel quarantine was made mandatory for international arrivals by air (*Government of Canada, 2021*). The downstream effects of border porosity could have been better mitigated by earlier widespread availability of rapid antigen tests to inform quarantine duration (*Centers for Disease Control and Prevention, 2020*), and through increased resources for contact tracing, cluster response, and compliance oversight. While the majority of importations were inferred to have been into Quebec and Ontario, this could be partially attributable to differences in sequence generation and deposition. Based on provincial population sizes alone, one would expect more importations into Ontario and Quebec (populations of 14.8 M and 8.6 M) than into British Columbia (5.2 M), Alberta (4.5 M), or other provinces and territories with less than 1.5 M (*Canada Statistics, 2021b*).

During the first wave, early sublineages had the opportunity to become established and resulted in large transmission chains driving COVID-19 burden, consistent with findings from the UK (*du Plessis et al., 2021*). However, the second wave was mostly driven by cases from newly seeded sublineages.

We accounted for potential confounding due to recent sublineages having been monitored for less time by stratifying sublineages by whether they were active or extinct in the previous 2 months; and by restricting the data to collection dates 3 months prior to the data download, we accounted for delay in depositing sequence data for more recent sublineages. Differences remained between early and late sublineages' evolutionary dynamics in the presence of VOCs with heightened transmissibility and the roll-out of vaccines. The earliest introductions were typified by high sublineage lifespans (i.e. duration of transmission), which manifested as an initial increase in the mean time since importation, corroborating that outbreak control via sublineage detection and contract tracing improved over the course of the first wave. Sublineage TMRCAs typically lag importation by several days, but could precede them if multiple infections with subsequently sampled descendants occurred during travel. While accounting for sampling lag between symptoms and sample collection would get closer to the importation date, this assumes symptom onset occurred during travel, which might not always be the case. In contrast, if we assume that infected travellers who sparked domestic transmission chains were tested upon or soon after arrival, then this lag would be zero to a couple of days. If an infectious traveller was not sampled, but gave rise to an outbreak that was later sampled, then an additional lag must be considered regarding the length and number generations of unsampled viral predecessors since importation. A TMRCA could precede importation if the sublineage index case infected multiple people during travel. Where travel history is available, incorporating internal nodes or branch states for travel location would help to inform the timing as well as geographic origins of importations (*Hong et al., 2021*). In light of these additional sources of uncertainty, we abstained from applying an importation lag to TMRCAs, and emphasize their interpretation as the estimated time of the recent common ancestor of all sampled descendants.

Case burden in the second wave was primarily due to sublineages introduced in the summer of 2020 amid low prevalence, despite a low number of importations. This suggests that ongoing seeding events impacted viral persistence prior to the arrival of VOCs with increased transmissibility. Considering the global and temporal diversity of the SARS-CoV-2 pandemic, it is difficult to predict what variant may be introduced and subsequently disperse through a local population. While broad and longstanding restrictions against non-essential international travel is not necessarily an advisable policy in light of economic impacts, swift and stringent travel bans towards locations harbouring a high frequency of a VOC not yet identified domestically should be seriously considered to reduce the probability of seeding multiple, simultaneous outbreaks and potentially overwhelming the health-care system. Dynamic travel bans can be detrimental from a socioeconomic perspective; therefore their implementation must be weighed relative to other non-pharmaceutical interventions to reduce transmission. As illustrated by Omicron, perhaps pervasive spread is inevitable with the most transmissible and immune evasive variants. However, reducing the number of importations sparking domestic outbreaks grants jurisdictions more time to prepare for the inevitable by ramping up vaccinations, scaling up testing and contact tracing, allocating primary care resources, and implementing non-pharmaceutical interventions. These advanced planning steps also serve to minimize the economic harm due to work absenteeism and supply chain disruption.

Ancestral trait reconstruction is sensitive to sampling bias as ancestral node states are more likely to reflect states predominant among tips (*Baele et al., 2017*; *Hill et al., 2021*; *Lemey et al., 2009*). By subsampling sequences proportionally to monthly relative case counts, a prior is assumed towards the probability of importation from each geography. While subsampling proportionally to cases reduces representation of geographies with the highest sequence contributions, it does not directly increase representation of geographies with sparse sequences, which may have relatively underestimated relative migration rates. Furthermore, countries and provinces differed in the extent of their contact tracing, testing, and case reporting, which adds uncertainty in the ascertainment rate underlying observed cases. Hospitalizations or COVID-related deaths could be used to impute or deconvolute the true number of people infected (*Huisman et al., 2021*), but this requires strong assumptions about the hospitalization and case fatality rate, both functions of the virulence of viral variants and prior immunological exposure of the population. In the present analysis, we make the simplifying assumption that case detection rates are comparable across geographies, justifying the use of case contributions to inform the subsampling process, however we recognize that additional data clarifying reason for testing could improve these estimates. The subsampling sensitivity analysis revealed how subsampling iteratively reduced overrepresentation in available provincial sequences and improves

the correlation between cases and sequences. Across subsampling strategies, the relative spatiotemporal trends were robust, although exact estimates were variable, particularly for smaller values.

Incorporating individuals' travel history or flight volume data could help to improve representation of undersampled geographies (*Lemey et al., 2020*; *du Plessis et al., 2021*). However, Bayesian phylogeography remains limited in its scalability despite improved efficiencies in the memory required (*Didelot et al., 2018*; *du Plessis et al., 2021*), and they are still limited to thousands, rather than tens or hundreds of thousand sequences. Therefore, the maximum likelihood approach that we have applied offers scalability, while taking into consideration the uncertainty of sampling bias by including multiple bootstraps and subsampling strategies. Stratified analyses of well-supported (sub)lineages in future analyses will facilitate direct comparisons of Bayesian and maximum likelihood phylogeography methodologies towards estimating importation rates.

Low sequence representation can lead to underestimates of total introductions if neither index case nor descendants were sampled, underestimates of sublineage size if not all descendants were sampled, and similarly, overestimates of the proportion of singletons, which may have been from unsampled transmission chains. Extrapolating an upper estimate of introductions is challenging in the absence of additional data. Clean genomes available in Canada prior to 1 March 2021 represented 4.2% of confirmed diagnoses (and 3.2% when 75% of Canadian sequences retained). Diagnoses were estimated to represent about 9% of total cases in Canada up to September 2020, while other geographies ranged from 5% in Italy to 99% in Qatar (*Noh and Danuser, 2021*). The probability of a case being detected is affected by geography (sociodemographic structure, testing capacity, and recommendations), by individual (age, contact-traced, political beliefs, co-morbidities), and by lineage (symptom severity, infectivity profile). Reason for sequencing is not always random – it could be for an outbreak investigation or to confirm VOC identity – and it varies over time by jurisdiction. As more sequences are generated and made available, we expect more descendants of previously identified sublineages than travellers or their recent contacts harbouring new sublineages or singletons. When sequencing efforts or resources are lower, travellers are a more efficient use of resources if prevalence is higher abroad than domestically, increasing the travel bias. Thus, importations do not scale linearly with sequence representation. In theory, the upper limit of importations by province could be estimated by adjusting for monthly sequence representation, case ascertainment rate, outbreak bias (ratio of probabilities of testing given infected for random versus outbreak-linked), and travel bias (ratio of probabilities of testing given infected for domestic versus travelling populations) over time, stratified by geography. More consistent inclusion of the reason for sequencing and testing in the publicly available metadata could facilitate better estimates of the extent of travel-related and outbreak-related bias. Additionally, prospective cohort studies or seroprevalence studies would ameliorate our estimate of the case ascertainment fraction.

These analyses shed light upon the natural epidemiological history of SARS-CoV-2 in the context of public health interventions and exemplify a sublineage-based method of genomic surveillance. Sharing viral genome sequences linked with the time and place of sampling in a timely manner is of utmost importance for epidemic surveillance of new and already described variants, analyses to support contact tracing, and inference of SARS-CoV-2 importation dynamics.

# Materials and methods
## Timeline of COVID-19 in Canada

The spatiotemporal dynamics of average daily COVID-19 diagnoses and SARS-CoV-2 sequences available on GISAID in Canada during the first and second waves prior to 1 March 2021 were summarized in the context of the Oxford stringency index, key epidemic events, and national-level interventions (*Figure 1*). Number of new cases by Canadian province and territory over time was obtained from the *Public Health Agency of Canada, 2021*. The distribution of sampled Pango lineages over time was summarized as raw daily sequences and frequencies (*Figure 1—figure supplement 1*). Dates of national-level COVID-19 interventions were obtained from the *Canadian Institute for Health Information, 2021* and key epidemiological events were obtained from a summary published by the Canadian Press via the National Post (*Press, 2021*). The Oxford Stringency Index for Canada was obtained from the Oxford COVID-19 Government Response Tracker *Hale et al., 2021*; it is a composite metric of national-level containment and closure policies including school and workplace closures, cancellations

of public events, gathering restrictions, stay at home requirements, restrictions on internal movement, and international travel controls. Maps were made using NAD83 datum shapefiles from the *Canada Statistics, 2019* Census (*Canada Statistics, 2019*) and from the *United States Census Bureau, 2019*.

## Sequence cleaning

1,999,711 SARS-CoV-2 sequences were downloaded from GISAID with metadata on 17 June 2021, of which 63,645 were sampled in Canada (*Khare et al., 2021*; *Shu and McCauley, 2017*). Contributing and submitting laboratories of all subsampled sequences are acknowledged in *Supplementary file 1*. An initial cleaning step was applied to remove sequences if they were listed in the Nextstrain exclude list on the day of data download (n=6272) (*Bedford, 2021*; *Hadfield et al., 2018*), duplicate entries (n=790), from a non-human host (n=2057), environmental samples (n=965), likely to contain sequencing errors based on previous temporal analyses (n=2) (*Rambaut, 2020*), or had incomplete sample collection dates (n=60,076). Canadian sequences with incomplete collection dates (n=19,915) were retained unless they lacked the month of collection (n=83). Remaining sequences were aligned to the Wuhan-Hu-1 reference sequence (GenBank MN908947.3) using the viralMSA python wrapper invoking minimap2 (*Li, 2018*; *Moshiri and Robinson, 2021*). A subsequent cleaning step was applied to remove sequences that contained more than 20% ambiguous sites (n=2370) or 10% gaps (n=123,174). Pango lineage designations were assigned using pangolin v3.0.5, pangoLEARN release 2021-06-05 (*O'Toole et al., 2021*). To focus on the first and second waves, as well as limit the right-censoring effect of sequence deposition delays described elsewhere (*Kalia et al., 2021*), the cleaned alignment of 1,825,297 sequences (60,143 Canadian) was filtered to only include sequences with collection dates preceding 1 March 2021, resulting in 953,242 sequences (36,736 Canadian). Problematic sites (v5) were masked in the alignment prior to phylogenetic inference (*de Maio, 2020*).

## Subsampling sequences

Although in an earlier pre-print version of this analysis using data downloaded in February 2021, all 9657 clean Canadian sequences were sampled alongside 40,333 global sequences (*McLaughlin et al., 2021*), in this updated analysis, 36,736 clean Canadian sequences in the study period necessitated subsampling Canadian in addition to global sequences. Alternative subsampling strategies with varying representation of Canadian sequences were compared in a subsampling sensitivity analysis described below. To maximize our ability to detect introductions, 75% (n=27,552) of available Canadian sequences were sampled in the primary analysis along with 22,448 global sequences, up to a total of 50,000 sequences for computational feasibility (*Figure 1—figure supplement 2*). Confirmed COVID-19 diagnoses by province from the Public Health Agency of Canada were aggregated by month in order to calculate each province's contribution to monthly Canadian new diagnoses (PHAC, 2021). Canadian sequences were sampled with probabilities scaled by proportional case contributions, distributed as evenly as possible across all months. Global sequences were subsampled similarly. Country-specific daily new diagnoses from the R package coronavirus (*Krispin and Byrnes, 2020*) were aggregated by month, and used to calculate the proportion of total new diagnoses in each country among all new global diagnoses, which was applied as a sampling probability for sequences from that month. Since there were relatively few sequences available until March 2020, all clean sequences in the preceding months were included and the remainder of sequences were sampled equally among subsequent months (*Supplementary file 3a*). Subsampling was repeated for 10 bootstraps with replacement. Estimates are reported as the mean across bootstraps and 95% confidence intervals were calculated using the t-distribution, $\mu \pm t \times \sigma/\sqrt{n}$.

## Phylogeographic inference

Approximate maximum likelihood phylogenies were inferred for every subsampled and masked alignment using FastTree v2.2.1 with a generalized time-reversible substitution model (*Price et al., 2010*). Trees were outgroup-rooted on Wuhan-hu-1 using R package ape (*Paradis and Schliep, 2019*). A linear regression of evolutionary distance over time was fit using TempEst v1.5 (*Rambaut et al., 2016*) and temporal outliers were excluded if residuals were more than three standard deviations from the mean or pendant edges represented more than 12 mutations (*Figure 2—figure supplement 1*; *Supplementary file 3c*). Phylogenies were time-scaled using LSD2 within IQ-TREE 2.1.2 (*Minh et al., 2020*; *To et al., 2016*), specifying a lognormal relaxed clock with 0.2 relative variance, a generalized

time-reversible substitution model with gamma rate variation, invariant sites, three discrete rate categories (*Lanfear, 2020*), Wuhan-Hu-1 (GenBank MN908947.3) as outgroup, and branch lengths resampled 50 times to calculate confidence intervals on dates (*Supplementary file 3d*).

Maximum likelihood discrete ancestral state reconstruction was applied on bifurcating time-scaled trees with randomly resolved polytomies, where tip state was designated either Canadian province or country of sampling, using the ace function of R package ape (*Paradis and Schliep, 2019*). Since Iran and Italy were known to have low sequence representation in early 2020 (*Worobey et al., 2020*), seven Canadian sequences with travel history to those regions prior to June 2020 had their tip state re-assigned to their country of travel. Travel history was unavailable for the large majority of sequences. The highest likelihood state was pulled for each internal node. Canadian sublineages were designated where a Canadian internal node was preceded by a non-Canadian internal node, signifying an international introduction resulting in onward transmission. Singletons were defined as sampled Canadian sequences with an international parental origin and no sampled descendants. Countries were grouped by continent if they were the likely origin for an average of fewer than five sublineages and the Canadian provinces Nova Scotia, New Brunswick, and Newfoundland and Labrador were merged into the Maritimes. Sublineage importation rate was summarized as the 7-day rolling mean of importations per week by global region, province of introduction, and lineage. The sum of sublineages and singletons represents a lower limit for the total number of introductions.

## Sublineage characterization

Sublineages were named by the most common Pango lineage among unique descendants, followed by the suffix 'can' and a numeric based on the order of first Canadian sample dates among each Pango lineage. Early sublineages, such as B.1.can1, may consist of a mixture of Pango lineages and in some cases contained nested sublineages representing re-introductions. If a sequence was a descendant of multiple nested sublineages, they were only considered a unique descendant of the most recently introduced sublineage. Bootstraps differed in their sublineage identification and nomenclature; therefore, we summarized the highest likelihood bootstrap from the 75% subsampling strategy (*Figure 3*). Sublineages' TMRCA, the approximate date of the first transmission event resulting in a sampled descendant following viral introduction, were estimated as the mean with 95% confidence intervals.

Sublineages were compared in regards to size, detection lag, and transmission lifespan over time (*Figure 4—figure supplement 2*). Sublineage size was defined as the number of uniquely sampled descendants within a subtree defined by the Canadian introductory node. A negative binomial model of sublineage size by TMRCA was generated, adjusted by whether a sublineage was active (had any sampled cases in the previous 2 months) or extinct, using the R package MASS (*Venables and Ripley, 2002*). Since the data was downloaded on 15 June, but restricted to dates before 1 March, the effect of sequence deposition delay on sublineage lifespan was partially mitigated. Models additionally adjusting by province were evaluated but unsupported by likelihood ratio tests and Bayesian information criterion (BIC). Sublineage detection lag was estimated as the number of days between the TMRCA and the first Canadian sample collection. Differences in the detection lags between provinces were compared using a non-parametric Kruskal-Wallis rank sum test, followed by a pairwise Dunn's rank sum test with Bonferroni p-value adjustment (*Dinno, 2017*). Multiple linear regression was used to evaluate whether detection lag was associated with TMRCA. The inclusion of province as a confounder was evaluated using likelihood ratio tests and BIC. Sublineage transmission lifespan, that is, the calendar duration of a transmission chain's persistence, was defined as the number of days from the TMRCA to the most recent Canadian descendant's sample date. A negative binomial model was applied to evaluate whether sublineage lifespan was significantly reduced over time. To account for sublineages' differing sizes, the days since importation was estimated for all Canadian sublineage descendants, calculated as the number of days between sampling date and the TMRCA. A 14-day centred rolling mean with 95% confidence interval was calculated to evaluate the changes in the average age since importation of SARS-CoV-2 sequences.

## Transmission sources of Canadian sequences

The parental geography of all Canadian tips in the phylogeny was estimated, which represent all sampled transmission events in which a Canadian was the recipient, including sublineage descendants and singletons. Transmission sources across Canadian tips were categorized as within-province,

between-province, USA, or other international (*Figure 5*). The proportional contribution from all international sources to sampled transmission events in April 2020, following enactment of stringent travel restrictions, was mapped as a chloropleth (*Figure 5—figure supplement 1*). Between-province transmission was quantified among Canadian tips with a parent from another Canadian province and was reported as the relative number of sampled transmission events where a given province was the source or the recipient.

## Subsampling sensitivity analysis

To evaluate the effect of differing levels of Canadian sequence inclusion relative to global representation, a subsampling sensitivity analysis was conducted comparing four subsampling strategies with 25% (n=9184), 50% (n=18,368), 75% (n=27,552), or 100% (n=36,736) of available clean Canadian sequences subsampled proportionally to monthly provincial new diagnoses and the remainder of global sequences subsampled proportionally to monthly global new diagnoses up to 50,000 total (*Figure 6*). The temporal distribution of sequences sampled each month was equalized as best as possible, with all sparse months' sequences included and the remainder of sequences distributed equally across months (*Supplementary file 3a*). For each strategy, 10 bootstraps were sampled with replacement. All 40 subsampled alignments were analysed through the remainder of the phylogeographic pipeline as described previously. The sensitivity of the results was assessed in regards to the 7-day rolling average of number of sublineages imported into provinces over time and the relative contributions of global origins in each wave.

The mean and 95% confidence intervals were calculated across bootstraps using the t-distribution and compared across strategies for metrics including number of sublineages, singletons, and total imports overall and by location pair; proportion of all imports that were sublineages; and sublineage importation rates over time. The sensitivity of the sublineage characterization models including detection lag over time, number of descendants over time stratified by active, and sublineage lifespan stratified by active were also compared across strategies.

## R packages

R packages used in the cleaning, subsampling, and phylogenetic analysis included ape 5.4–1 (*Paradis et al., 2004*), Biostrings 3.1.3 (*Pagès et al., 2020*), phytools 0.7–70 (*Revell, 2012*), phangorn 2.5.5 (*Schliep et al., 2017*), tidyverse 1.3.1 (*Wickham et al., 2019*), coronavirus 0.3.0.9000 (*Krispin and Byrnes, 2020*), lubridate 1.7.9.2 (*Grolemund and Wickham, 2011*), zoo 1.8–8 (*Zeileis and Grothendieck, 2005*), RColorBrewer 1.1–2 (*Neuwirth, 2014*), cowplot 1.1.1 (*Wilke, 2020*), ggstance 0.3.5 (*Henry et al., 2020*), ggalluvial 0.12.3 (*Brunson, 2020*), ggridges 0.5.3 (*Wilke, 2021*), ggtree 2.2.4 (*Yu et al., 2016*), ggplotify 0.0.5 (*Yu, 2020*), ggrepel 0.9.1 (*Slowikowski, 2021*), MASS 7.3–53 (*Venables and Ripley, 2002*), and treemapify 2.5.5 (*Wilkins, 2021*). Additional R packages used to generate the maps included rgeos 0.5–5 (*Bivand and Rundel, 2018b*), maptools 1.0–2 (*Bivand and Lewin-Koh, 2018a*), ggsn 0.5.0 (*Baquero, 2017*), broom 0.7.6 (*Couch, 2021*), and rgdal 1.5–18 (*Bivand et al., 2017*).

## Acknowledgements

We are very grateful to all the contributing and submitting laboratories within Canada and internationally that generously deposited sequences and metadata on GISAID (*Supplementary file 1*). We are especially thankful to the members of the Canadian COVID-19 Genomics Network (CanCOGeN) Consortium (*Supplementary file 2*) and the Canadian Public Health Laboratory Network (CPHLN) for their contributions towards publicly available data. Funding Statement: AM was supported by a Canadian Institutes for Health Research (CIHR) Doctoral grant and a Natural Sciences and Engineering Research Council of Canada (NSERC) CREATE scholarship. RLM was supported by an NSERC CREATE scholarship. GM was supported by the Liber Ero Fellowship Programme. MW was supported by the David and Lucile Packard Foundation. AFYP was supported by a CIHR Project Grant PJT-156178. JBJ was supported by Genome Canada BCB 287PHY grant, an operating grant from the CIHR Coronavirus Rapid Response Programme number 440371, and a CIHR variant of concern supplement. The British Columbia Centre for Excellence in HIV/AIDS also provided support.

## Additional information

### Competing interests

Canadian COVID-19 Genomics Network (CanCOGen) Consortium: The other authors declare that no competing interests exist.

### Funding

| Funder | Grant reference number | Author |
| --- | --- | --- |
| Canadian Institutes for Health Research | Canada Graduate Scholarship Doctoral Awards | Angela McLaughlin |
| National Sciences and Engineering Research Council of Canada | CREATE Award | Angela McLaughlin Rachel L Miller |
| Liber Ero Foundation | Fellowship | Gideon J Mordecai |
| David and Lucile Packard Foundation | Grant | Michael Worobey |
| Canadian Institutes for Health Research | Project Grant PJT-156178 | Art FY Poon |
| Genome Canada | BCB 287PHY Grant | Jeffrey B Joy |
| Canadian Institutes for Health Research | Coronavirus Rapid Response ProgrammeCoronavirus Rapid Response Operating Grant 440371 | Jeffrey B Joy |
| Canadian Institutes for Health Research | Variant of Concern Supplement Grant | Jeffrey B Joy |
| British Columbia Centre for Excellence in HIV/AIDS | Funding | Angela McLaughlin Vincent Montoya Rachel L Miller Jeffrey B Joy |

The funders had no role in study design, data collection and interpretation, or the decision to submit the work for publication.

### Author contributions

Angela McLaughlin, Conceptualization, Data curation, Software, Formal analysis, Funding acquisition, Validation, Investigation, Visualization, Methodology, Writing – original draft, Project administration, Writing – review and editing, AM conceived of the project, conducted most of the data cleaning, analysis, and visualization, and wrote the original draft of the manuscript; Vincent Montoya, Data curation, Formal analysis, Methodology, Writing – review and editing, VM helped to develop methodologies for data cleaning, alignment, and tree building; Rachel L Miller, Visualization, Writing – review and editing, RLM contributed towards improving the graphics; Gideon J Mordecai, Visualization, Methodology, Writing – review and editing, GJM provided feedback on the phylogeographic methods and graphics, as well as provided citations; Canadian COVID-19 Genomics Network (CanCOGen) Consortium, Funding acquisition, Writing – review and editing, Initial feedback on earlier versions of the analysis; Michael Worobey, Conceptualization, Supervision, Funding acquisition, Visualization, Methodology, Writing – review and editing, MW contributed suggestions on estimating uncertainty, visualizations, and interpretations; Art FY Poon, Data curation, Supervision, Visualization, Methodology, Writing – review and editing; Jeffrey B Joy, Conceptualization, Resources, Data curation, Software, Supervision, Funding acquisition, Validation, Visualization, Methodology, Writing – original draft, Project administration, Writing – review and editing, JBJ contributed to refining the research questions

### Author ORCIDs

Angela McLaughlin http://orcid.org/0000-0001-5606-9080
Art FY Poon http://orcid.org/0000-0003-3779-154X
Jeffrey B Joy http://orcid.org/0000-0002-7013-1482

Decision letter and Author response
Decision letter https://doi.org/10.7554/eLife.73896.sa1
Author response https://doi.org/10.7554/eLife.73896.sa2

---

## Additional files

### Supplementary files
• Supplementary file 1. An acknowledgment of contributing laboratories who generated and uploaded viral genetic sequences and metadata to Global Initiative on Sharing All Influenza Data (GISAID).

• Supplementary file 2. Canadian COVID-19 Genomics Network (CanCOGen) consortium membership.

• Supplementary file 3. Supplemental tables from subsampling sensitivity analysis.

• Transparent reporting form

### Data availability
All the viral genetic sequences analyzed in this study were sourced from the Global initiative on sharing all influenza data (GISAID) coronavirus (CoV) database and are subject to the GISAID EpiFlu Database Access Agreement. Within this agreement, we are disallowed to distribute data to any third party other than Authorized Users as contemplated by this Agreement. In lieu, a table of accession IDs was provided in Supplementary file 1, along with an acknowledgement of all originating and submitting laboratories. Submission of this table is also a condition of use of the data under the terms and conditions of GISAID. Individuals who would like to become Authorized Users may submit an application https://www.gisaid.org/registration/register/ and agree to the Database Access Agreement. Full and subsampled alignments can be shared to Authorized Users upon request. The scripts used for data curation, inferences, analyses, and visualization are available with a reproducible example at https://github.com/AngMcL/sars-cov-2_canada_2020 (copy archived at swh:1:rev:1cb49fa43c752b1ce1003c52e43b5c47d3b6789f).

The following previously published dataset was used:

| Author(s) | Year | Dataset title | Dataset URL | Database and Identifier |
|---|---|---|---|---|
| Shu Y, McCauley J | 2017 | GISAID: Global initiative on sharing all influenza data - coronovius (CoV) | https://doi.org/10.2807/1560-7917.ES.2017.22.13.30494 | GISAID, 10.2807/1560-7917.ES.2017.22.13.30494 |

---

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
