## [Editor Report]

This study provides important observations about the transmission of SARS-CoV-2 lineages within Canada and the importation of lineages into Canada over the first year of the COVID-19 pandemic. This information is critical for understanding SARS-CoV-2 evolution and epidemiology, including the potential impacts of travel restrictions.

---

## [Decision Letter]

**Decision letter after peer review:**

Thank you for submitting your article "Early introductions of SARS-CoV-2 sublineages into Canada drove the 2020 epidemic" for consideration by *eLife*. Your article has been reviewed by 3 peer reviewers, one of whom is a member of our Board of Reviewing Editors, and the evaluation has been overseen by David Serwadda as the Senior Editor. The following individual involved in review of your submission has agreed to reveal their identity: Bernardo Gutierrez (Reviewer #3).

Essential revisions:

All reviewers thought this study represented a potentially valuable contribution. They also all expressed concern about the impact of potentially biased sampling on the conclusions. We recognize that this is a general challenge in the field, and not all papers in it have necessarily faced this level of scrutiny. Because the conclusions of this paper are quantitative, it seems important to quantify the uncertainty (in this case, to model misspecification/sampling bias) accurately.

The reviewers outline nuances in their individual commentaries below, but there are effectively three general issues:

1) It seems particularly important to understand (e.g., as a sensitivity analysis) the impact of not downsampling the samples from Canada.

2) Reviewers also asked whether the case-based scaling of sequences is appropriate. This relates to a more general problem raised by the highly uneven sampling by country and region. Downsampling so that no region is overrepresented with respect to its relative infection load would leave zero samples for analysis, since so many areas are missing sequences entirely. Perhaps again performing some kind of sensitivity analysis or referencing a more thorough investigation of this problem, how does uneven representation impact estimated migration rates?

3) The reviewers were also concerned with the estimate of the total number of introductions.

Effectively, all the reviews are requesting a more careful treatment of potential problems with data collection (it is not random!), both in the analyses and in the text.

*Reviewer #1 (Recommendations for the authors):*

A few other statistical questions are raised in the recommendations for the authors below.

Statistics/methodology

1. p. 5, l. 3: Clearer logic in the "Subsampling" section would help. What is the justification for subsampling proportional to case number, which is obviously quite biased by surveillance effort, perhaps even more biased than the number of sequences? Why not subsample proportional to countries' population size or excess mortality, adjusted for age? It would be helpful to describe too how much subsampling occurred and that it was necessary for computational reasons. What kinds of biases are still present? For instance, it would be good to point out somewhere the regions/periods that are so poorly sampled (e.g., in many lower income countries) that the inference could be biased to favor endemic transmission, assuming my intuition is correct.

2. p. 8, ll. 6-8: Could you explain the logic of how the total number of introductions was inferred given the estimated rate at which cases were sequenced (and cases detected)? Should the estimated number of introductions scale linearly?

3. p. 12, ll. 6-9 (Figure 4C): Does this model properly adjust for the typical delay between sampling members of a sublineage? Sublineages with more recent tMRCA should have fewer descendants simply because less time has passed. It is not clear the stratification takes care of this. There might also be a decrease in the number of descendants with time if the total number of sublineages is increasing and surveillance does not scale with sublineage richness. The text reads "To elucidate the relative contributions of early and late sublineages,….," but it is unclear exactly what contribution is being measured here.

4. p. 10, ll. 10-11 (Figure 4D): I think we could see an increase in days since importation over time even if more recently imported lineages were generally more common ("dominant"). The maximum possible age of the oldest lineage is increasing with time. The sentence says this "suggests" that early introductions "increasingly dominated" the epidemic. Maybe it would be easier to demonstrate this through specific examples or a measure of persistence time and a proxy of prevalence.

5. p. 18, ll. 6-8: "Sequences from individuals with travel history to Iran or Italy before June 2020 were recategorized as having been sampled in the country of travel." Why was this not done for individuals with travel history from other countries?

Interpretation

1. Abstract: "Rapid implementation of stringent border controls and quarantine could have diminished the Canadian COVID-19 burden by curtailing the spread of early introductions." I question how much stringent border controls, unless defined very severely as prohibiting the introduction of, e.g., more than ten lineages, could have diminished the COVID-19 burden. Once any lineage is established and transmitting in the community, additional introductions have negligible impact on prevalence or the "final size" of an epidemic. Quarantine, by reducing the effective reproductive number, can reduce burden. I worry that lack of precision on the impact of introductions on burden can lead to the sort of knee-jerk travel bans we see with Omicron.

2. p. 3, ll. 7-8: "can illuminate"? Deciphering the rates of imported and domestic transmission is not guaranteed.

3. p. 8, l. 8: A 42% infection detection rate (as cases) struck me as rather high. The cited reference appears to be for something else.

4. p. 10, ll. 1-3: What is the purpose of mentioning the number of global descendants when it is known to be such an underestimate?

5. Figure 4. It would be useful to reference 4A and 4B specifically in the text.

6. p. 15, ll. 4-6: How can we separate the impacts of behavioral changes (NPIs) from seasonality on sublineage size and duration?

7. p. 15, ll. 22-24: Where is the mandatory 14-day quarantine coming from? Given the incubation period and duration of shedding, I know many who think quarantine should be shorter (if testing were not feasible). With sensitive testing, there is no reason to keep people past a week if they are consistently negative. I think it's important not to imply the traditional durations used for quarantine are especially grounded or optimized.

Potential enhancements

It would be interesting to see how the importation rates from different countries correlate with air travel to/from those countries. I realize this is difficult given connecting flights, but perhaps the data exist somewhere.

p. 16, ll. 11-14: Ideally these nonstationary sampling probabilities by province (and really for all countries) would be built directly into the likelihood.

*Reviewer #2 (Recommendations for the authors):*

1. Line 17-21 argue genomic epidemiology is useful to track and characterize VOI/VOC, but the sequences analyzed in this manuscript are largely pre-VOC so I think it makes sense to re-focus and expand this discussion to include commentary on the importance of evaluating the success of public health interventions, travel restrictions, etc.

2. I think outlining a hypothesis at the end of the intro is useful, but as it is currently written the hypothesis is very long and difficult to understand. Can the authors restate and clarify the hypothesis into a single sentence?

3. As the authors discuss, their results and those of all genomic epidemiological studies are very sensitive to sequence quality, availability, distribution, and sampling paradigms. They reduce the impact of unequal global sequencing efforts by subsampling global sequences proportionally to their monthly case counts, excluding Canadian sequences. I understand the rationale here, but worry that inclusion of all available Canadian sequences (which are not proportional to case count trends as is clear in Figure 1D and 1E) skewed their results. Perhaps one way to assess the robustness of the results would be to repeat the main analyses using a subsampled (proportion to case counts) Canadian dataset and compare results? Relatedly, even case counts could be skewed depending on testing availability and accuracy -- could the authors include a brief commentary on the availability, accessibility, and capacity of testing centers in Canada during the first year of the pandemic?

- it is also clear to see in Figure 1C and 1F that US sequences are overrepresented in this dataset compared to case counts -- do the authors think this is inflating the proportion of US imports? (I agree it is clear most imports came from the US)

- the Alberta and Maritimes outbreaks in spring of 2020 appear very striking (large outbreaks with lots of person-to-person spread and very few descendants) in Figure. However, this interpretation is fraught when you consider Figure 1E and 1D because there are lots of available sequences during this time compared to the total case count vs fall of 2020 where there is actually a higher case burden and very few available sequences. I think many readers will get hung up on this so maybe the authors could address it briefly in the text?

- While the color scheme is lovely, it is difficult to distinguish between some provinces particularly in Figure 2

4. Lines 6-10. I think it is a big leap to estimate the total possible introductions based on sequences representative of 1.1% of total cases.

5. It might be very informative to take a more careful look at rates of viral spread (R0) over time in addition to patterns of viral importation. This could help contextualize the effect of public health interventions on domestic spread. The authors bring up quarantine, contact tracing, testing, and individual behavior change in the discussion -- do the authors think the rate of international imports is the best measure to assess these interventions? I think R0 would be just as informative!

6. I am skeptical the results on reductions in sublineage size, increasing tMRC, and perhaps even sources of domestic transmission will persist if the Canadian sequences are subsampled proportionally to case counts. Can the authors please assess the robustness of these results against a subsampled dataset?

7. I agree with and appreciate the authors' discussion of the importance data sharing, including a limited metadata set to aid in the rapid interpretation and use of sequence data for important public health decisions.

8. Although this is a stylistic preference, I believe a cleaned up version of Figure 7 with the Canada's key public health interventions and COVID-19 outbreaks along with a map of Canada's provinces (already a supplemental) and case counts during the study period would be very useful to include as an early, primary figure.

*Reviewer #3 (Recommendations for the authors):*

My main concern lies on the explicit naming of sources of importation following the application of an epidemiologically-based approach to downsample the publicly available genome sequences from GISAID. While it is mentioned that ten subsamples were generated, it is not clear to me that the ancestral state reconstruction was evaluated on these ten subsamples and how the results compare between them. The uncertainty of the ASR itself was incorporated for the fixed phylogeny through an ML approach (ape::ace()), but I am missing the analysis of the robustness of the results across different subsamples.

Furthermore, even if these analyses where performed, it is unclear whether a different subsampling approach could lead to the identification of a different profile of source countries, and if so, which subsampling approach would be more accurate. The accuracy of the identified countries of origin (and therefore accuracy of the proposed downsampling approach) would be validated if a complementary data source showed similar patterns: an expected importation index (EII ~ Incidence * no. of travellers to Canada), self-reported travel histories of patients with confirmed COVID-19, etc. If these complementary data sources are unavailable, I would recommend reducing the emphasis on the identification of countries of origin (or constraining it to regions where you're confident that the monthly reported cases are reliable – the USA would be an obvious example of this), or adding a clear discussion on these limitations. I would argue that this goes beyond the already mentioned idea that downsampling does not allow to increase the numbers of sequences from some unsampled times/locations (page 16, lines 5-20), and that it can also reduce the probability of phylogenetically identifying true importations from countries with low case prevalence.

On other topics: I am unsure about the estimation of total number of introductions presented in page 8 (lines 6-10). Do you have any particular reason why you would expect that importations scale linearly with sampling/representation? Given the overall dynamics, wouldn't it be more probable that higher sampling intensities would add unsampled sequences to the larger sublineages in the country (proportional to the size of the lineages) as well as uncovering unsampled importations? The rationale behind these calculations requires some more detailed explanations.

Regarding the domestic transmission, I was wondering if you had access to finer geographical resolution beyond province-level? Provinces are particularly large geographical areas, and providing some insights (rather than a full, high-resolution phylogeographic re-analysis of all the data) into the sampling within provinces could add to the manuscript. This is particularly the case for the analysis of international importations to Quebec and Ontario: where these sublineages first detected in large urban areas or not necessarily? Is the sampling within provinces representative of the cases reported in urban and rural areas? Are there any differential dynamics between larger urban areas and more remote locations?

Finally, it is quite interesting to see that the peak of TMRCAs nicely coincide with the maximum stringency on March 21 (Figure 4A,B) – even more so given that the TMRCA of a sublineage is more likely representing something akin to the first transmission event within said sublineage rather than the importation event per se. It would be interesting if the TMRCAs could be contextualised to show their distribution after accounting for sampling lag (i.e., the time between symptom onset and sample collection). This could be estimated from sequence metadata or approximated from non-genomic epidemiological data, but it would give a better idea of the times during which the infections that lead to these sublineages started.

[Editors' note: further revisions were suggested prior to acceptance, as described below.]

Thank you for resubmitting your work entitled "Genomic epidemiology of the first two waves of SARS-CoV-2 in Canada" for further consideration by *eLife*. Your revised article has been evaluated by Sara Sawyer (Senior Editor) and a Reviewing Editor.

The manuscript has been dramatically improved---the sensitivity analyses are immensely helpful---but there are some remaining issues that need to be addressed, as outlined below. The reviewing editor is concerned that some of the policy suggestions do not follow the evidence presented.

1. Abstract: "… suggesting travel restrictions and quarantine must be sustained to fully curtail COVID-19 burden." I am guessing most readers will likely interpret "to fully curtail COVID-19 burden" as to get the burden near zero. Is this what the authors intended? It affects the logic. The results here show that travel restrictions were usually associated with at least several importations per week. Mathematically, we know that the burden of COVID-19 does not really depend on the number of importations if R ~ 1 or more---all you need is at least one endemic lineage to maintain the prevalence/burden dictated by R. Maybe the authors are referring to conditions when R<1 and travel restrictions are even greater than what they have been historically, but this is unclear. It is also unclear precisely what the authors mean by "quarantine." Usually, it refers to the isolation of contacts of potential cases (not the cases themselves), and sometimes it refers more generally to lockdowns. Neither policy seems particularly worth calling out here. Why not say "sustained interventions to lower transmission (R<1)"? It seems unjustified for this study to single out quarantine (either def) as the necessary complementary intervention when so many other approaches have been shown more effective. I'm dwelling on the wording because these kinds of conclusions, especially in an abstract, can have an outsized impact on policy.

2. Abstract: "…restrictions that reduce the probability of importations are most effective during periods of low domestic prevalence and low or waning immunity." Waning immunity means R is rising, which, all things equal reduces extinction probabilities and increases the success of invading lineages (e.g., Otto and Whitlock 1997). Without more precision about R and how effectiveness is being defined, it's not clear restrictions should be more impactful under these conditions. I do see the positive correlation between "periods of waning immunity" and possible ban-assisted extinction, but only because immunity is most often waning when there has recently been an epidemic, and the recently boosted immune protection is relatively high. I worry again about the implications of these claims, which are not exactly supported by the research here.

3. Relatedly, the main text describes evaluating the relative contribution of (reducing) importations to (reducing) *burden* (e.g., ll. 190-192, ll. 605-607, 691-701), but this analysis is not really performed here. Such analysis would require careful estimation of R and prevalence and a good enough transmission model to evaluate extinction probabilities and outbreak sizes (in the case of R<1) over time, and it would have to account for clonal interference. Put differently, the fact that imported lineages caused x% of cases does not mean that the case count would've been (100-x)% without those importations. The fact that imported lineages displaced resident lineages does not mean the resident lineages could not have caused comparable infections had the imported lineages been stopped at the border, assuming the imported lineages had no fitness differences. Imperfect travel bans that allow some lineages to invade can at best slightly (at measured rates) change the timing of an epidemic (ll. 704-707), but usually not its size or severity. The authors write, "Although blind travel bans may not be beneficial from a socioeconomic perspective, ultimately governments need to protect citizens and dynamic travel bans are one of the few tools available." This is surprising considering the number of broader tools at our disposal (e.g., rapid testing, vaccination, ventilation/filtration) that can slow importations AND endemic spread, as well as the observed ineffectiveness of travel bans at stopping spread outside of islands and China. The authors IMO have not demonstrated how much bans could delay importation and allow time to "plan", or if investments to reduce transmission generally are more effective. I suggest these claims be carefully reconsidered.

I apologize for the wordiness. I would be happy to bolster these points (maybe with math) if they are unclear. I am concerned that too much is being extrapolated about the possible and actual epidemiological impacts of travel bans from the measured rates of importation.

You may wish to consider some suggestions to improve readability from another reviewer:

4. In line 424, I'm not entirely sure what the authors mean by "the first wave of singletons had a higher amplitude than the second wave". By 'amplitude', do they refer to the range of dates when singletons were identified between both waves? Is this relative to a feature of the importations of sublineages (given that this sentence talks about this comparison between sublineage and singleton dynamics)? Some clarification would be helpful.

5. The Discussion addresses a lot of the reviewer comments thoroughly and systematically, which was very helpful, but could be streamlined for the readers. I think that, for example, the two paragraphs that go from line 711 to line 741 could be reduced to a single paragraph discussing 1. sampling bias in phylogeography, 2. the specific effects of overrepresentation of some locations and how they addressed it with subsampling and the sensitivity analyses, 3. effects of upsampled lineages and under-sampled locations and how this can theoretically be overcome, 4. the broader challenges of scaling up Bayesian phylogenetics (this last point could be a separate paragraph, but unrelated to sampling bias).

---

## [Author Response]

Essential revisions:All reviewers thought this study represented a potentially valuable contribution. They also all expressed concern about the impact of potentially biased sampling on the conclusions. We recognize that this is a general challenge in the field, and not all papers in it have necessarily faced this level of scrutiny. Because the conclusions of this paper are quantitative, it seems important to quantify the uncertainty (in this case, to model misspecification/sampling bias) accurately.

We are very grateful to the reviewers and editors for their time in reviewing our manuscript and for the many helpful comments, suggestions and clarifying questions, which we have incorporated in the revised manuscript. Most notably, we have assessed the impact of sample bias on importation rate estimates though an empirical subsampling sensitivity analysis. Additionally, we have updated the entire analysis with a more recently downloaded dataset, which included about four times the number of Canadian sequences, and an expanded study period up until March 2021 to include the entirety of the second wave. To accommodate more Canadian sequences, we have extended the global subsampling strategy to subsample Canadian sequences proportionally to provincial case counts. To address the issues of potential sampling bias on relative importations into provinces over time, we explored multiple subsampling strategies in which 25%, 50%, 75%, or 100% of Canadian sequences were retained, with the remainder of sequences from the global set up to 50,000 total. We found that under all the subsampling strategies, our conclusions were robust with regards to the impact of the travel restrictions, the temporal dynamics of importation rates, and the relative contributions of geographies towards sublineages in the first and second waves.

The reviewers outline nuances in their individual commentaries below, but there are effectively three general issues:1) It seems particularly important to understand (e.g., as a sensitivity analysis) the impact of not downsampling the samples from Canada.

We thank the reviewers for identifying the importance of this limitation. We have addressed the impact of subsampling the Canadian sequences relative to global context sequences in a sensitivity analysis.

Methods

“Subsampling sensitivity analysis

To evaluate the effect of differing levels of Canadian sequence inclusion relative to global representation, a subsampling sensitivity analysis was conducted comparing four subsampling strategies with 25% (n = 9,184), 50% (n = 18,368), 75% (n = 27,552), or 100% (n = 36,736) of available clean Canadian sequences subsampled proportionally to monthly provincial new diagnoses and the remainder of global sequences subsampled proportionally to monthly global new diagnoses up to 50,000 total (Figure 6). […] The sensitivity of the sublineage characterization models including detection lag over time, number of descendants over time stratified by active, and sublineage lifespan stratified by active were also compared across strategies.”

Results

“Subsampling sensitivity analysis

To render phylogenetic inferences computationally feasible, reduce overrepresentation of geographies with more sequences per case, and evaluate uncertainty in the effect of sampling on importation rates, sequences were subsampled proportionally to either provinces or countries’ contributions to monthly case counts in Canada and globally for multiple bootstraps. […] Across all strategies, there were significant reductions of sublineage size and lifespan over time.”

2) Reviewers also asked whether the case-based scaling of sequences is appropriate. This relates to a more general problem raised by the highly uneven sampling by country and region. Downsampling so that no region is overrepresented with respect to its relative infection load would leave zero samples for analysis, since so many areas are missing sequences entirely. Perhaps again performing some kind of sensitivity analysis or referencing a more thorough investigation of this problem, how does uneven representation impact estimated migration rates?

We thank the reviewers for acknowledging this key issue in interpreting phylogeographic migration rates. We believe this issue has been partially addressed with the empirical subsampling sensitivity analysis. Sampling bias has been further elaborated upon in the Discussion, with additional citations on the sensitivity of discrete trait analysis to sampling bias.

“Ancestral trait reconstruction is sensitive to sampling bias as ancestral node states are more likely to reflect states predominant among tips (Baele et al., 2016; Hill et al., 2021; Lemey et al., 2009). […] Stratified analyses of well-supported (sub)lineages in future analyses will facilitate direct comparisons of Bayesian and maximum likelihood phylogeography methods employed here.”

3) The reviewers were also concerned with the estimate of the total number of introductions.

We gratefully acknowledge the reviewers’ concern regarding the rough extrapolation of an upper limit on the total number of introductions. Rather than estimate the upper limit, we have elaborated upon the type of data required to attempt this with any shred of confidence.

“Low sequence representation can lead to underestimates of total introductions if neither index case or descendants were sampled, underestimates of sublineage size if not all descendants were sampled, and similarly, overestimates of the proportion of singletons, which may have been from unsampled transmission chains. […] Additionally, prospective cohort studies or seroprevalence studies would ameliorate our estimate of the case ascertainment fraction.”

Effectively, all the reviews are requesting a more careful treatment of potential problems with data collection (it is not random!), both in the analyses and in the text.Reviewer #1 (Recommendations for the authors):A few other statistical questions are raised in the recommendations for the authors below.Statistics/methodology1. p. 5, l. 3: Clearer logic in the "Subsampling" section would help. What is the justification for subsampling proportional to case number, which is obviously quite biased by surveillance effort, perhaps even more biased than the number of sequences? Why not subsample proportional to countries' population size or excess mortality, adjusted for age?

We acknowledge the reviewer’s concern regarding the effect of differences in surveillance upon case proportional subsampling, however we maintain our perspective that the extent of bias is greater among available sequences than it is among detected cases. The supplementary figures comparing sequences per case demonstrate the utility of this subsampling method in improving the correlation of cases to sequences. While it would be possible to sample proportionally to population size, this exerts a prior belief that infected cases should be equally distributed globally and that a larger population necessarily results in a larger risk at any given time. An example corroborating this would be China, which contributed most cases in January and February 2020, but very few relative cases in late 2020 and early 2021, despite a large population size. Alternatively, COVID deaths or excess mortality could have been used, but are subject to biases of their own. Definitions and identification of COVID deaths differed over time and space, and while excess mortality would account for missed deaths, it would also include excess deaths due to other indirect causes (delayed surgeries due to COVID, for instance). We have added to the Discussion elaborating upon the justification of this method and the sensitivity analyses we conducted regarding the assumptions.

“Ancestral trait reconstruction is sensitive to sampling bias as ancestral node states are more likely to reflect states predominant among tips (Baele et al., 2016; Hill et al., 2021; Lemey et al., 2009). […] The subsampling sensitivity analysis revealed how subsampling iteratively reduced overrepresentation in available provincial sequences and improves the correlation between cases and sequences.”

It would be helpful to describe too how much subsampling occurred and that it was necessary for computational reasons. What kinds of biases are still present? For instance, it would be good to point out somewhere the regions/periods that are so poorly sampled (e.g., in many lower income countries) that the inference could be biased to favor endemic transmission, assuming my intuition is correct.

We have clarified the extent of subsampling by geography by sampling strategy.

The general effect of lingering sampling biases was elaborated upon in the Discussion (see excerpt above) and more specifically in regards to the overrepresentation of the USA:

“Although the USA was highly represented in all of our subsamples as a result of its large contribution to COVID-19 cases in 2020 and high sequence availability during early months, our results suggest a greater effect than due to sampling alone. On average, the USA sequences represented 28.9% (28.7 – 29.2%) of total international sequences, yet accounted for 46.3% (44.0 – 48.7%) of all sublineages and 57.7% (55.6 – 59.8%) of singletons. Upon maximizing the number of Canadian sequences in the analysis, where global sequence representation was more normalized but less comprehensive, the USA sequences represented fewer of the international sequences (25.8%, 25.6 – 26.1%) and still accounted for 38.4% (37.0 – 39.8%) of sublineages and 46.4% (44.6 – 48.3%) of singletons.”

The reviewer posited that the inference could be biased to favor endemic transmission, by which we assume they meant the inference would be more likely to detect local or domestic transmission sources than international sources of sampled Canadian viral diversity. This is possible in instances where there was an absence of sequences sharing a recent common ancestor with Canadian sequences from the true sublineage origin location, which itself was seeded by a previous ancestor with descendants in Canada. In the subsampling sensitivity analysis, where Canadian sequences were differentially represented in the data, the proportion of transmission attributable to international sources was comparable, although with fewer total estimated importations when too few Canadian sequences were included (25% strategy) or too few global sequences were included (100% strategy). In the 75% strategy, there were nearly the same number of Canadian and global context sequences, which if we assume a well-mixed population, confers a similar probability of detecting domestic versus international transmission sources. In reality, the global sample is not randomly drawn from a well-mixed viral pool, despite our best efforts. Naturally, we will miss detecting an importation from a geography with no sequences available. We comment upon the potential of advances in methodology by Phillipe Lemey and colleagues towards incorporating unsampled nodes in phylogeographic inference to mitigating this issue.

“Incorporating individuals’ travel history or flight volume data could help to improve representation of undersampled geographies (Lemey et al., 2020; du Plessis et al., 2021). However, Bayesian phylogeography remains limited in its scalability despite improved efficiencies in the memory required (Didelot et al., 2018; du Plessis et al., 2021), and they are still limited to thousands, rather than tens or hundreds of thousand sequences. Therefore, the maximum likelihood approach that we have applied offers scalability, while taking into consideration the uncertainty of sampling bias by including ten bootstraps for four subsampling strategies. Random sampling permits an exploration of the effect of country and Canadian province representation over time on the results. In comparing strategies, the relative spatiotemporal trends were robust, while raw estimates were more variable, as were smaller rate estimates. Stratified analyses of well-supported (sub)lineages in future analyses will facilitate direct comparisons of Bayesian and maximum likelihood phylogeography methods employed here.”

2. p. 8, ll. 6-8: Could you explain the logic of how the total number of introductions was inferred given the estimated rate at which cases were sequenced (and cases detected)? Should the estimated number of introductions scale linearly?

We gratefully acknowledge the reviewer’s concern regarding the rough extrapolation of an upper limit on the total number of introductions. Rather than estimate the upper limit, we have elaborated upon the type of data required to attempt this with any shred of confidence.

“Low sequence representation can lead to underestimates of total introductions if neither index case or descendants were sampled, underestimates of sublineage size if not all descendants were sampled, and similarly, overestimates of the proportion of singletons, which may have been from unsampled transmission chains. […] Additionally, prospective cohort studies or seroprevalence studies would ameliorate our estimate of the case ascertainment fraction.”

3. p. 12, ll. 6-9 (Figure 4C): Does this model properly adjust for the typical delay between sampling members of a sublineage? Sublineages with more recent tMRCA should have fewer descendants simply because less time has passed. It is not clear the stratification takes care of this. There might also be a decrease in the number of descendants with time if the total number of sublineages is increasing and surveillance does not scale with sublineage richness. The text reads "To elucidate the relative contributions of early and late sublineages,….," but it is unclear exactly what contribution is being measured here.

In the updated analysis, we included data up to March 2021 that had been deposited by June 2021. This permitted us to include the entire first and second waves, reducing the effect of differential delays in observing descendants. While the stratification of sublineages on the basis of having any sampled descendants in the previous two months does not entirely adjust for the delay in deposition, it improves our confidence in asserting trends among extinct sublineages. In regards to whether we might observe a decrease in the number of descendants with time if the total number of sublineages is increasing but surveillance does not scale with sublineage richness, this assumes that the testing capacity is surpassed. Until the testing capacity in a given region is reached and surpassed, the proportion of a sublineage’s descendants that are tested and sequenced should remain relatively constant, notwithstanding differences in contact tracing implementation. Although testing capacity was exceeded later in 2021, this did not affect data from the first two waves as much.

“During the first wave, early sublineages had the opportunity to become established and resulted in large transmission chains driving COVID-19 burden, consistent with findings from the UK (du Plessis et al., 2021). However, the second wave was mostly driven by cases from newly seeded sublineages. We accounted for potential confounding due to recent sublineages having been monitored for less time by stratifying sublineages by whether they were active or extinct in the previous two months; and by restricting the data to collection dates three months prior to the data download, we accounted for delay in depositing sequence data for more recent sublineages. Differences remained between early and late sublineages’ evolutionary dynamics in the presence of VOCs with heightened transmissibility and the roll-out of vaccines.”

4. p. 10, ll. 10-11 (Figure 4D): I think we could see an increase in days since importation over time even if more recently imported lineages were generally more common ("dominant"). The maximum possible age of the oldest lineage is increasing with time. The sentence says this "suggests" that early introductions "increasingly dominated" the epidemic. Maybe it would be easier to demonstrate this through specific examples or a measure of persistence time and a proxy of prevalence.

We agree with the reviewer that this requires clarification and validation. Similar to persistence time or duration of transmission, we have modelled sublineage lifespan, the time period between the TMRCA and last sample date, and stratified sublineages activity based on whether there were any samples in the last two months or if they appear to have gone extinct. By updating the analysis with data available up to 15 June 2021, but only investigating data sampled prior to 1 March, the effect of deposition delay on right truncation was reduced.

5. p. 18, ll. 6-8: "Sequences from individuals with travel history to Iran or Italy before June 2020 were recategorized as having been sampled in the country of travel." Why was this not done for individuals with travel history from other countries?

There were very few Canadian sequences with travel histories available (n=10). Besides sequences associated with Iran or Italy, these included one of each with travel history from Wuhan, Hong Kong, and an international cruise ship. Although we could have re-assigned the state of Canadian sequences with travel history to Wuhan and Hong Kong, the majority of sequences available during January and February 2020 were from China anyways. Furthermore, re-assigning tip state prevents us from identifying these sequences as introductions, so it is only a practice we would recommend for the most severely undersampled countries, which Iran and Italy were during the early pandemic.

Interpretation1. Abstract: "Rapid implementation of stringent border controls and quarantine could have diminished the Canadian COVID-19 burden by curtailing the spread of early introductions." I question how much stringent border controls, unless defined very severely as prohibiting the introduction of, e.g., more than ten lineages, could have diminished the COVID-19 burden. Once any lineage is established and transmitting in the community, additional introductions have negligible impact on prevalence or the "final size" of an epidemic. Quarantine, by reducing the effective reproductive number, can reduce burden. I worry that lack of precision on the impact of introductions on burden can lead to the sort of knee-jerk travel bans we see with Omicron.

Thank you for your perspective on this subject, which remains an active discussion. We have elaborated upon this dialogue in the Discussion.

“The burden of cases in the second wave was primarily due to sublineages introduced in the summer of 2020, despite a low number of importations and case burden at that time. […] These advanced planning steps also serve to minimize the economic harm due to work absenteeism and supply chain disruption.”

2. p. 3, ll. 7-8: "can illuminate"? Deciphering the rates of imported and domestic transmission is not guaranteed.

Thank you for this suggestion. We have changed the wording.

“Studying evolutionary relationships between SARS-CoV-2 sequences over time and space allows estimation of the relative contributions of international and domestic transmission in association with public health interventions.”

3. p. 8, l. 8: A 42% infection detection rate (as cases) struck me as rather high. The cited reference appears to be for something else.

Thank you for this point. We have identified a more appropriate and robust reference for the case ascertainment rate in Canada in 2020.

“Clean genomes available in Canada prior to 1 March 2021 represented 4.2% of confirmed diagnoses (and 3.2% when 75% of Canadian sequences retained). Diagnoses were estimated to represent about 9% of total cases in Canada up to September 2020, while other geographies ranged from 5% in Italy to 99% in Qatar (Noh & Danuser, 2021).”

4. p. 10, ll. 1-3: What is the purpose of mentioning the number of global descendants when it is known to be such an underestimate?

Although it will naturally be an underestimate as a consequence of subsampling and incomplete case ascertainment, we have included it as a way of showing that migration flows both ways and that Canadian sublineages have been exported out of the country. While absolute numbers cannot be extracted, the relative representation of countries among sampled descendants can be interpreted with some uncertainty.

5. Figure 4. It would be useful to reference 4A and 4B specifically in the text.

Please note that the figure numbering has changed during our revisions. What were previously figure 4A and B are now Figure 5A and Figure 5—figure supplement 1. They are referenced in the text here:

“Following the entry restriction for foreign nationals in mid-March, there was a reduction, but not an elimination, of the proportion of transmission events attributable to the USA or other international sources across all provinces (Figure 5A). Prior to the restrictions, provinces had a mean of 26% of transmission events with any international origins, ranging from 17% (14-20%) in Saskatchewan to 33% (31-35%) in British Columbia. In April 2020, following travel restrictions, all provinces had decreased proportions of international transmission sources, yet Ontario and Quebec retained relatively high mean number of transmission events attributable to international sources, at 115 (105-125) and 95 (82-109) (Figure 5—figure supplement 1). This may suggest slower implementation or compliance with quarantine guidelines in these provinces.”

6. p. 15, ll. 4-6: How can we separate the impacts of behavioral changes (NPIs) from seasonality on sublineage size and duration?

We agree with the reviewer that seasonality is an important confounding factor to consider in associating differences in transmission with the implementation of NPIs or behavioral changes –factors such as temperature, humidity, and socializing inside in poorly ventilated areas can all affect transmission, and thus sublineage size and duration. In this 2020 analysis, it was not possible to disentangle these effects over the span of one year. If might be tempting to compare to 2021, but the emergence and domination of VOCs make them hard to compare. Within 2020, one might consider comparing the effect of a given NPI, such as restrictions on international travel, on sublineage size and duration in different seasons in Northern and Southern hemisphere countries. However, the challenge here is that countries different in the timing and strength of their NPIs and had different dominant sublineages circulating at any given time. Although the relationship with sublineage size and duration with time may be confounded by seasonality, the reduction in importations during Spring 2020 was too rapid and drastic to have occurred as a result of differences in season.

7. p. 15, ll. 22-24: Where is the mandatory 14-day quarantine coming from? Given the incubation period and duration of shedding, I know many who think quarantine should be shorter (if testing were not feasible). With sensitive testing, there is no reason to keep people past a week if they are consistently negative. I think it's important not to imply the traditional durations used for quarantine are especially grounded or optimized.

We have reworded the discussion to not necessarily recommend a 14-day quarantine.

“The downstream effects of border porosity could have been better mitigated by earlier widespread availability of rapid antigen tests to inform quarantine duration (CDC, 2020), and through increased resources for contact tracing, cluster response, and compliance oversight.”

Potential enhancementsIt would be interesting to see how the importation rates from different countries correlate with air travel to/from those countries. I realize this is difficult given connecting flights, but perhaps the data exist somewhere.

We agree with the reviewer that this would be an interesting and challenging comparison of expected versus observed importation rates. This is similar to what was undertaken by du Plessis et al., where both prevalence and flight volume were considered towards an estimated Importation Intensity. If we had access to volumes of arrival flights or number of passengers into Canada by country for 2020, we could make this comparison. If there were more importations than expected, this may suggest systematic differences in quarantine or testing oversight based on where people arrived from. However, flight data does not incorporate land crossings, which we expect accounted for a large number of importations from the USA. We discussed the potential role played by travel across the land border towards importations from the USA.

“It was foreseeable that the USA would be the largest source of SARS-CoV-2 importations into Canada given its high COVID-19 prevalence throughout 2020 as well as the fact that these nations share the world’s longest land border, spanning 8,890 km. Overall international arrivals into Canada declined 77.8% from 96.8M in 2019 to 19.7M in 2020 (Statistics Canada, 2021a). However, the number of truck drivers and other crew members only declined by 24.8%, accounting for 49% of all international arrivals after April 2020 (of whom 93% were truck drivers). Truck drivers supported the supply chain throughout COVID-19 disruptions, yet in doing so, may have inadvertently facilitated additional SARS-CoV-2 importations from the USA. International arrivals of Canadians and non-residents in 2020 were low, but not negligible, with 14.6M and 5.1M arrivals respectively in 2020 (Statistics Canada, 2021a). While essential workers were exempt from quarantine, all others were mandated a self-enforced 14-day quarantine upon re-entry. It was not until 22 February 2021 that a 3-day hotel quarantine was made mandatory for international arrivals by air (Government of Canada, 2021).”

p. 16, ll. 11-14: Ideally these nonstationary sampling probabilities by province (and really for all countries) would be built directly into the likelihood.

We agree with the reviewer that it would be interesting to incorporate sampling probabilities or proportions into either the likelihood of the internal node state or into the importation rates derived from them. We considered this, however decided against it for the following reasons. Pragmatically, the software used for ancestral character estimation is not flexible towards specifying priors on migration rates. We considered adjusting importation rates post hoc with the time-varying sampling probabilities by province and by country of origin, but doing so is subject to the same biases as our previous extrapolation of the upper estimate of the total number of introductions. We do not know the extent of travel bias or outbreak bias in the sequences, nor the case ascertainment rate, therefore we cannot simply assume that sequences were a random sample nor that importation rates scale linearly with available sequences. We will consider this suggestion for a future analysis using a software that can better accommodate extensions to the likelihood function.

Reviewer #2 (Recommendations for the authors):1. Line 17-21 argue genomic epidemiology is useful to track and characterize VOI/VOC, but the sequences analyzed in this manuscript are largely pre-VOC so I think it makes sense to re-focus and expand this discussion to include commentary on the importance of evaluating the success of public health interventions, travel restrictions, etc.

We thank the reviewer for this recommendation and have changed the introduction to focus more on the precedence and importance of evaluating effectiveness of public health interventions.

“Characterization of viral importations over time can also clarify the effectiveness of public health interventions by associating inflection points in importations with drastic changes in policies such as international travel restrictions (Magalis et al., 2020). […] Evaluating travel restrictions’ effectiveness towards reducing burden could inform the stringency and timing of future public health interventions.”

The discussion also elaborated upon pragmatic policy changes that could be informed by these evaluations:

“The burden of cases in the second wave was primarily due to sublineages introduced in the summer of 2020, despite a low number of importations and case burden at that time. […] These advanced planning steps also serve to minimize the economic harm due to work absenteeism and supply chain disruption.”

2. I think outlining a hypothesis at the end of the intro is useful, but as it is currently written the hypothesis is very long and difficult to understand. Can the authors restate and clarify the hypothesis into a single sentence?

Thank you for the suggestion. We have distilled the scope into a single hypothesis:

“We tested the hypothesis that international travel restrictions enacted in March 2020 effectively reduced international importations of SARS-CoV-2 into Canada, yet ongoing introductions contributed to COVID-19 persistence into early 2021, exacerbated by highly transmissible B.1.1.7 and other VOC sublineages.”

3. As the authors discuss, their results and those of all genomic epidemiological studies are very sensitive to sequence quality, availability, distribution, and sampling paradigms. They reduce the impact of unequal global sequencing efforts by subsampling global sequences proportionally to their monthly case counts, excluding Canadian sequences. I understand the rationale here, but worry that inclusion of all available Canadian sequences (which are not proportional to case count trends as is clear in Figure 1D and 1E) skewed their results. Perhaps one way to assess the robustness of the results would be to repeat the main analyses using a subsampled (proportion to case counts) Canadian dataset and compare results? Relatedly, even case counts could be skewed depending on testing availability and accuracy -- could the authors include a brief commentary on the availability, accessibility, and capacity of testing centers in Canada during the first year of the pandemic?

We fully agree with the reviewer and have undertaken sensitivity analyses to address these concerns, as cited in text above. Additionally, in regards to the potential bias towards identifying USA importations across subsampling strategies, we added the following remark.

“Although the USA was highly represented in all of our subsamples as a result of its large contribution to COVID-19 cases in 2020 and high sequence availability during early months, our results suggest a greater effect than due to sampling alone. On average, the USA sequences represented 28.9% (28.7 – 29.2%) of total international sequences, yet accounted for 46.3% (44.0 – 48.7%) of all sublineages and 57.7% (55.6 – 59.8%) of singletons. Upon maximizing the number of Canadian sequences in the analysis, where global sequence representation was more normalized but less comprehensive, the USA sequences represented fewer of the international sequences (25.8%, 25.6 – 26.1%) and still accounted for 38.4% (37.0 – 39.8%) of sublineages and 46.4% (44.6 – 48.3%) of singletons.”

– The Alberta and Maritimes outbreaks in spring of 2020 appear very striking (large outbreaks with lots of person-to-person spread and very few descendants) in Figure. However, this interpretation is fraught when you consider Figure 1E and 1D because there are lots of available sequences during this time compared to the total case count vs fall of 2020 where there is actually a higher case burden and very few available sequences. I think many readers will get hung up on this so maybe the authors could address it briefly in the text?

We agree that the relative availability of sequences in the first and second waves is a limitation for several provinces, particularly Alberta. In the updated analysis, there was an increased representation of the second wave, improving this issue somewhat.

4. Lines 6-10. I think it is a big leap to estimate the total possible introductions based on sequences representative of 1.1% of total cases.

We agree with the reviewer that this upper estimate required further consideration of travel-related bias, outbreak-related bias, and case ascertainment rates. Since these values are not known across provinces, we have removed this upper estimate and replaced it with a discussion of the type of data needed to extrapolate the true number of introductions.

“Low sequence representation can lead to underestimates of total introductions if neither index case or descendants were sampled, underestimates of sublineage size if not all descendants were sampled, and similarly, overestimates of the proportion of singletons, which may have been from unsampled transmission chains. […] Additionally, prospective cohort studies or seroprevalence studies would ameliorate our estimate of the case ascertainment fraction.”

5. It might be very informative to take a more careful look at rates of viral spread (R0) over time in addition to patterns of viral importation. This could help contextualize the effect of public health interventions on domestic spread. The authors bring up quarantine, contact tracing, testing, and individual behavior change in the discussion -- do the authors think the rate of international imports is the best measure to assess these interventions? I think R0 would be just as informative!

We thank the reviewer for pointing out that showing differences in domestic viral spread over time would strengthen the conclusions regarding the effectiveness of NPIs. In regards to what is the best measure of the effectiveness of travel restrictions, we maintain that looking at importations over time is paramount. We experimented with modelling the effective reproduction number over time using national and provincial incidence, as well as sublineage descendants sampled over time, however there was considerable uncertainty in these estimates due to uncertainty in the serial interval and when occurrences were infrequent. We are developing more refined methods of deconvoluting the incidence data to improve these estimates for a follow-up publication. Instead, to address the reviewer’s request for a more thorough characterization of domestic spread, we have added a figure showing the distribution of sampled Canadian descendants of key sublineages (Figure 2B) and respective subtrees (Figure 2—figure supplement 1).

6. I am skeptical the results on reductions in sublineage size, increasing tMRC, and perhaps even sources of domestic transmission will persist if the Canadian sequences are subsampled proportionally to case counts. Can the authors please assess the robustness of these results against a subsampled dataset?

We have undertaken a Canada sequence subsampling sensitivity analysis and evaluated the effects on sublineage size, TMRCA, and relative sources of domestic transmission analysis. Please see subsampling sensitivity analysis.

7. I agree with and appreciate the authors' discussion of the importance data sharing, including a limited metadata set to aid in the rapid interpretation and use of sequence data for important public health decisions.

We thank the reviewer for sharing our opinion of how data sharing augments our collective knowledge of the epidemic and better positions us to respond effectively.

8. Although this is a stylistic preference, I believe a cleaned up version of Figure 7 with the Canada's key public health interventions and COVID-19 outbreaks along with a map of Canada's provinces (already a supplemental) and case counts during the study period would be very useful to include as an early, primary figure.

Thank you – we agreed with the reviewer’s recommendation to move the summary of interventions, stringency, cases, and sequences to be the first primary figure, as it gives an overview of the epidemic and sets the stage for the analysis. We also added the Canadian map in place of the legend in Figure 1A to provide geographic context for provincial names.

Reviewer #3 (Recommendations for the authors):As mentioned in the Public Review, my main concern lies on the explicit naming of sources of importation following the application of an epidemiologically-based approach to downsample the publicly available genome sequences from GISAID. While it is mentioned that ten subsamples were generated, it is not clear to me that the ancestral state reconstruction was evaluated on these ten subsamples and how the results compare between them. The uncertainty of the ASR itself was incorporated for the fixed phylogeny through an ML approach (ape::ace()), but I am missing the analysis of the robustness of the results across different subsamples.

We thank the reviewer for noting that we could better show the uncertainty across bootstraps. All of the reported estimates in the Results are the mean across ten bootstraps. Including uncertainty in every figure was not feasible as there were already many features in the plot, which we did not want to detract attention from. In the subsampling sensitivity analysis undertaken, the uncertainty across bootstraps for each subsampling strategy was shown more explicitly.

Furthermore, even if these analyses where performed, it is unclear whether a different subsampling approach could lead to the identification of a different profile of source countries, and if so, which subsampling approach would be more accurate. The accuracy of the identified countries of origin (and therefore accuracy of the proposed downsampling approach) would be validated if a complementary data source showed similar patterns: an expected importation index (EII ~ Incidence * no. of travellers to Canada), self-reported travel histories of patients with confirmed COVID-19, etc. If these complementary data sources are unavailable, I would recommend reducing the emphasis on the identification of countries of origin (or constraining it to regions where you're confident that the monthly reported cases are reliable – the USA would be an obvious example of this), or adding a clear discussion on these limitations. I would argue that this goes beyond the already mentioned idea that downsampling does not allow to increase the numbers of sequences from some unsampled times/locations (page 16, lines 5-20), and that it can also reduce the probability of phylogenetically identifying true importations from countries with low case prevalence.On other topics: I am unsure about the estimation of total number of introductions presented in page 8 (lines 6-10). Do you have any particular reason why you would expect that importations scale linearly with sampling/representation? Given the overall dynamics, wouldn't it be more probable that higher sampling intensities would add unsampled sequences to the larger sublineages in the country (proportional to the size of the lineages) as well as uncovering unsampled importations? The rationale behind these calculations requires some more detailed explanations.

We agree with the reviewer that the estimated total number of introductions was too simplistic and rife with assumptions. We have omitted this upper estimate and expanded upon why this is challenging to estimate in the discussion.

“Low sequence representation can lead to underestimates of total introductions if neither index case or descendants were sampled, underestimates of sublineage size if not all descendants were sampled, and similarly, overestimates of the proportion of singletons, which may have been from unsampled transmission chains. […] Additionally, prospective cohort studies or seroprevalence studies would ameliorate our estimate of the case ascertainment fraction.”

Regarding the domestic transmission, I was wondering if you had access to finer geographical resolution beyond province-level? Provinces are particularly large geographical areas, and providing some insights (rather than a full, high-resolution phylogeographic re-analysis of all the data) into the sampling within provinces could add to the manuscript. This is particularly the case for the analysis of international importations to Quebec and Ontario: where these sublineages first detected in large urban areas or not necessarily? Is the sampling within provinces representative of the cases reported in urban and rural areas? Are there any differential dynamics between larger urban areas and more remote locations?

Unfortunately, we did not have access to geography metadata for Canadian sequences at a finer scale than province. This information is not available in the public domain presumably for confidentiality reasons. Our group has collaborated with the BCCDC to model importations and transmission among the five health authorities in BC. We have also worked with Ontario public health colleagues to model relative contributions of international transmission using sequences available from over a dozen provincial health regions as part of an extensive sampling effort on 24 November 2020 for a point prevalence study. The manuscript for this work is still in preparation and we will take your suggestions into consideration therein, stratifying based on rural or urban geographic landscape. I would expect that generally urban centers act as major sources of viral transmission (based on density and travel proclivity) to more remote locations. However, we are not able to comment on this dynamic based on the data available in this study.

Finally, it is quite interesting to see that the peak of TMRCAs nicely coincide with the maximum stringency on March 21 (Figure 4A,B) – even more so given that the TMRCA of a sublineage is more likely representing something akin to the first transmission event within said sublineage rather than the importation event per se. It would be interesting if the TMRCAs could be contextualised to show their distribution after accounting for sampling lag (i.e., the time between symptom onset and sample collection). This could be estimated from sequence metadata or approximated from non-genomic epidemiological data, but it would give a better idea of the times during which the infections that lead to these sublineages started.

We thank the reviewer for distinguishing sublineage importation from TMRCAs. While we considered your suggestion to estimate and correct for sampling lag, we were hesitant to apply a fixed value or randomly draw from a distribution of lags in light of unknown variability described in the discussion below. Rather, we clarified the interpretation of sublineage TMRCAs.

“Sublineage TMRCAs typically lag importation by several days, but could precede them if multiple infections with subsequently sampled descendants occurred during travel. While accounting for sampling lag between symptoms and sample collection would get closer to the importation date, this assumes symptom onset occurred during travel, which might not always be the case. In contrast, if we assume that infected travellers who sparked domestic transmission chains were tested upon or near after arrival, then this lag would be zero to a couple days. If an infectious traveller was not sampled, but gave rise to an outbreak that was later sampled, then an additional lag must be considered regarding the length and number generations of unsampled viral predecessors since importation. A TMRCA could precede importation if the sublineage index case infected multiple people during travel. Where travel history is available, incorporating internal nodes or branch states for travel location would help to inform the timing as well as geographic origins of importations (Hong et al., 2021). In light of these additional sources of uncertainty, we abstained from applying an importation lag to TMRCAs, and emphasize their interpretation as the estimated time of the recent common ancestor of all sampled descendants.”

[Editors' note: further revisions were suggested prior to acceptance, as described below.]

The manuscript has been dramatically improved---the sensitivity analyses are immensely helpful---but there are some remaining issues that need to be addressed, as outlined below. The reviewing editor (also reviewer 1) is concerned that some of the policy suggestions do not follow the evidence presented.

We are highly grateful to the Senior Editor and Reviewing Editor for their comments and suggestions, which we have addressed below.

1. Abstract: "… suggesting travel restrictions and quarantine must be sustained to fully curtail COVID-19 burden." I am guessing most readers will likely interpret "to fully curtail COVID-19 burden" as to get the burden near zero. Is this what the authors intended? It affects the logic. The results here show that travel restrictions were usually associated with at least several importations per week. Mathematically, we know that the burden of COVID-19 does not really depend on the number of importations if R ~ 1 or more---all you need is at least one endemic lineage to maintain the prevalence/burden dictated by R. Maybe the authors are referring to conditions when R<1 and travel restrictions are even greater than what they have been historically, but this is unclear. It is also unclear precisely what the authors mean by "quarantine." Usually, it refers to the isolation of contacts of potential cases (not the cases themselves), and sometimes it refers more generally to lockdowns. Neither policy seems particularly worth calling out here. Why not say "sustained interventions to lower transmission (R<1)"? It seems unjustified for this study to single out quarantine (either def) as the necessary complementary intervention when so many other approaches have been shown more effective. I'm dwelling on the wording because these kinds of conclusions, especially in an abstract, can have an outsized impact on policy.

We thank the reviewer for identifying this statement as problematic. We have changed the wording as follows:

“Despite the drastic reduction in viral importations following travel restrictions, newly seeded sublineages in summer and fall 2020 contributed greatly to the persistence of COVID-19 cases in the second wave, highlighting the importance of sustained interventions to reduce transmission.”

2. Abstract: "…restrictions that reduce the probability of importations are most effective during periods of low domestic prevalence and low or waning immunity." Waning immunity means R is rising, which, all things equal reduces extinction probabilities and increases the success of invading lineages (e.g., Otto and Whitlock 1997). Without more precision about R and how effectiveness is being defined, it's not clear restrictions should be more impactful under these conditions. I do see the positive correlation between "periods of waning immunity" and possible ban-assisted extinction, but only because immunity is most often waning when there has recently been an epidemic, and the recently boosted immune protection is relatively high. I worry again about the implications of these claims, which are not exactly supported by the research here.

We agree that we have extrapolated our claims beyond what was supported by our findings, where we did not explicitly model changes in population-level immunity. This sentence has been modified accordingly:

“Although viral importations are nearly inevitable when global prevalence is high, with fewer importations there are fewer opportunities for novel variants to spark outbreaks in new communities or outcompete previously circulating lineages.”

3. Relatedly, the main text describes evaluating the relative contribution of (reducing) importations to (reducing) *burden* (e.g., ll. 190-192, ll. 605-607, 691-701), but this analysis is not really performed here. Such analysis would require careful estimation of R and prevalence and a good enough transmission model to evaluate extinction probabilities and outbreak sizes (in the case of R<1) over time, and it would have to account for clonal interference. Put differently, the fact that imported lineages caused x% of cases does not mean that the case count would've been (100-x)% without those importations. The fact that imported lineages displaced resident lineages does not mean the resident lineages could not have caused comparable infections had the imported lineages been stopped at the border, assuming the imported lineages had no fitness differences. Imperfect travel bans that allow some lineages to invade can at best slightly (at measured rates) change the timing of an epidemic (ll. 704-707), but usually not its size or severity. The authors write, "Although blind travel bans may not be beneficial from a socioeconomic perspective, ultimately governments need to protect citizens and dynamic travel bans are one of the few tools available." This is surprising considering the number of broader tools at our disposal (e.g., rapid testing, vaccination, ventilation/filtration) that can slow importations AND endemic spread, as well as the observed ineffectiveness of travel bans at stopping spread outside of islands and China. The authors IMO have not demonstrated how much bans could delay importation and allow time to "plan", or if investments to reduce transmission generally are more effective. I suggest these claims be carefully reconsidered.I apologize for the wordiness. I would be happy to bolster these points (maybe with math) if they are unclear. I am concerned that too much is being extrapolated about the possible and actual epidemiological impacts of travel bans from the measured rates of importation.

We appreciate the reviewer’s perspective on the semantics and interpretation of our findings. We have not asserted that the case count correlates directly with importations, but that each importation increases the chance of seeding a new community and the chance of introducing a more transmissible variant. Wording has been adjusted as follows:

ll. 190-192: “Evaluating travel restrictions’ effectiveness towards reducing importations could inform the stringency and timing of future public health interventions.”

ll. 605-607: “A more rapid and stringent public health response would have reduced the initial burden by preventing early sublineages from establishing widespread transmission chains that persisted throughout the first and second waves.”

ll. 691-705: “Case burden in the second wave was primarily due to sublineages introduced in the summer of 2020 amid low prevalence, despite a low number of importations. This suggests that ongoing seeding events impacted viral persistence prior to the arrival of VOCs with increased transmissibility. Considering the global and temporal diversity of the SARS-CoV-2 pandemic, it is difficult to predict what variant may be introduced and subsequently disperse through a local population. While broad and longstanding restrictions against non-essential international travel is not necessarily an advisable policy in light of economic impacts, swift and stringent travel bans towards locations harbouring a high frequency of a VOC not yet identified domestically should be seriously considered to reduce the probability of multiple, simultaneous outbreaks being seeded and potentially overwhelming the healthcare system.”

ll. 704-707: “Dynamic travel bans can be detrimental from a socioeconomic perspective; therefore their implementation must be weighed relative to other non-pharmaceutical interventions to reduce transmission.”

You may wish to consider some suggestions to improve readability from another reviewer:4. In line 424, I'm not entirely sure what the authors mean by "the first wave of singletons had a higher amplitude than the second wave". By 'amplitude', do they refer to the range of dates when singletons were identified between both waves? Is this relative to a feature of the importations of sublineages (given that this sentence talks about this comparison between sublineage and singleton dynamics)? Some clarification would be helpful.

Thank you for seeking clarification on the wording here. By amplitude, we meant the wave height of the weekly singletons introduction rate, or the total average number of singletons introduced per week. Wording changed as follows:

“Importation dynamics for singletons mostly mirrored sublineage trends (Figure 4—figure supplement 1; Figure 3—figure supplement 1), although the USA contributed relatively more towards singletons than sublineages in both waves, and the maximum singleton importation rate was higher in the first wave than the second wave.”

5. The Discussion addresses a lot of the reviewer comments thoroughly and systematically, which was very helpful, but could be streamlined for the readers. I think that, for example, the two paragraphs that go from line 711 to line 741 could be reduced to a single paragraph discussing 1. sampling bias in phylogeography, 2. the specific effects of overrepresentation of some locations and how they addressed it with subsampling and the sensitivity analyses, 3. effects of upsampled lineages and under-sampled locations and how this can theoretically be overcome, 4. the broader challenges of scaling up Bayesian phylogenetics (this last point could be a separate paragraph, but unrelated to sampling bias).

We are grateful for this feedback towards improving the flow of the Discussion and have modified the text as follows:

“Ancestral trait reconstruction is sensitive to sampling bias as ancestral node states are more likely to reflect states predominant among tips (Baele et al., 2016; Hill et al., 2021; Lemey et al., 2009). […] Stratified analyses of well-supported (sub)lineages in future analyses will facilitate direct comparisons of Bayesian and maximum likelihood phylogeography methodologies towards estimating importation rates.”